# Interpretable machine learning prediction of all-cause mortality

Wei Qiu [1], Hugh Chen[1], Ayse Berceste Dincer[1], Scott Lundberg[2], Matt Kaeberlein[3] & Su-In Lee[1✉]

## Abstract

**Background** Unlike linear models which are traditionally used to study all-cause mortality, complex machine learning models can capture non-linear interrelations and provide opportunities to identify unexplored risk factors. Explainable artificial intelligence can improve prediction accuracy over linear models and reveal great insights into outcomes like mortality. This paper comprehensively analyzes all-cause mortality by explaining complex machine learning models.

**Methods** We propose the IMPACT framework that uses XAI technique to explain a state-of-the-art tree ensemble mortality prediction model. We apply IMPACT to understand all-cause mortality for 1-, 3-, 5-, and 10-year follow-up times within the NHANES dataset, which contains 47,261 samples and 151 features.

**Results** We show that IMPACT models achieve higher accuracy than linear models and neural networks. Using IMPACT, we identify several overlooked risk factors and interaction effects. Furthermore, we identify relationships between laboratory features and mortality that may suggest adjusting established reference intervals. Finally, we develop highly accurate, efficient and interpretable mortality risk scores that can be used by medical professionals and individuals without medical expertise. We ensure generalizability by performing temporal validation of the mortality risk scores and external validation of important findings with the UK Biobank dataset.

**Conclusions** IMPACT's unique strength is the explainable prediction, which provides insights into the complex, non-linear relationships between mortality and features, while maintaining high accuracy. Our explainable risk scores could help individuals improve self-awareness of their health status and help clinicians identify patients with high risk. IMPACT takes a consequential step towards bringing contemporary developments in XAI to epidemiology.

## Plain language summary

This study identifies characteristics that will make a person more likely to die sooner than expected based on life expectancy for the population. We developed a computer program and applied it to information obtained about the characteristics and medical history of people from the USA. We identified previously unidentified characteristics that impact how likely it is someone will die sooner than expected, for example the circumference of the arm. We also identified combinations of characteristics that interact to increase the likelihood of death sooner than expected. By adding a person's characteristics to the program, the likelihood of death over the next 5 years can be calculated and characteristics identified that a person could modify to improve their health and reduce their chance of death during this period.

[1] Paul G. Allen School of Computer Science and Engineering, University of Washington, Seattle, WA, USA. [2] Microsoft Research, Redmond, WA, USA. [3] Department of Laboratory Medicine and Pathology, University of Washington, Seattle, WA, USA. ✉email: suinlee@cs.washington.edu

dentification of risk factors and prediction of all-cause mortality have long been important issues in epidemiology. Most prior studies identify risk factors using associations between each predictor and mortality[1–3]; only a few papers use multivariate linear models to predict mortality and identify risk factors[4,5]. In terms of prediction, a variety of linear mortality risk scores have been proposed to help characterize unhealthy individuals[6–8]. Although linear models have historically been popular because they are interpretable, modern complex machine learning (ML) models often achieve higher predictive accuracy because they can capture interactions among variables in addition to non-linear relationships (e.g., "U-shaped" relationships).

The field of artificial intelligence (AI) has seen considerable advances in supervised learning problems, which involve predicting an outcome variable (e.g., all-cause mortality) based on a set of features (e.g., individual-level characteristics). Notable applications of AI in healthcare include diabetic retinopathy detection in ophthalmology images[9], red blood cells classification[10], Alzheimer's disease prediction[11], lung cancer classification from histopathology images[12], and skin cancer classification[13]. Despite this progress, a major obstacle to the adoption of AI applications in healthcare is that many of them are considered "black box," which refers to their lack of interpretability. The inability to understand why a model makes a prediction is especially harmful in healthcare applications, where the patterns a model discovers can be even more important than its predictive accuracy. This is especially true in epidemiology, which aims to identify important variables to guide public health policy or detect risk predictors that warrant further study. To address this need, we turn to a variety of techniques to help us better understand complex ML models from the emerging area of explainable AI (XAI)[14–16].

In this paper, we present the IMPACT (Interpretable Machine learning Prediction of All-Cause morTality) framework (Fig. 1), which improves the interpretability of complex machine learning models for mortality prediction. We combine an accurate, complex ML model and a state-of-the-art XAI technique to predict all-cause mortality and conduct a systematic and integrated study of the relationships among many variables and all-cause mortality. We apply IMPACT to the NHANES (1999-2014) dataset to reveal important all-cause mortality findings. First, using explainable complex ML models rather than linear models, we identify risk predictors that are highly informative of future mortality. Second, our flexible models capture non-linear relationships, which provide more comprehensive information about the relationship between feature values and mortality risk: for example, the "inflection" points of risk predictors could provide a unique perspective of reference intervals that has consequential implications in public health. Third, understanding which features are the most important enables us to develop highly accurate, efficient (using less features) and interpretable mortality risk scores. Furthermore, the individualized explanation of risk scores can help users understand their most important risk factors and adjust their lifestyle. In Table 1, we compare the AUROCs between an existing mortality score or a biological age as reported in the original paper and the IMPACT-20 model tested for the corresponding follow-up time and age ranges in the NHANES dataset. We find that IMPACT risk scores (Supplementary Methods) have higher predictive power than popular mortality risk scores[5–8] and biological ages[17–20]. We ensure generalizability by performing temporal validation of the mortality risk scores and external validation of feature importances and important relationships with the UK Biobank dataset. All our results and risk scores are available on an interactive website (https://suinleelab.github.io/IMPACT) to encourage exploration of important risk predictors and support the use of interpretable

individual risk scores for individuals with and without medical expertise. The IMPACT framework can also be applied to other health outcomes and diseases to improve the predictive accuracy and interpretability of complex ML models in epidemiological studies.

## Methods
**Data cohorts.** This study primarily focuses on NHANES[21–23] (http://www.cdc.gov/nchs/nhanes.htm) data based on samples collected between 1999 and 2014. We include demographic, laboratory, examination, and questionnaire features that could be automatically matched across different NHANES cycles. The National Center for Health Statistics Research Ethics Review Board approved all NHANES protocols, and all participants gave informed consent. After data preprocessing (Supplementary Methods), 47,261 samples with 151 features (Supplementary Data 1) remain. Follow-up mortality data is provided from the date of survey participation through December 31, 2015. We predict all-cause mortality for two broad categories: (1) follow-up times of 1-year, 3-year, 5-year, and 10-year, and (2) age groups of < 40, 40–65, 65–80, and ≥ 80 years old. For mortality prediction with different follow-up times, we use samples of all ages. For different age groups, we fix the follow-up time to predict 5-year mortality and divide all samples for 5-year mortality prediction into four sets based on age. The dataset is randomly divided into training (80%) and testing (20%) sets. Demographic characteristics and sample size of the data for different tasks are shown in Supplementary Fig. 1 and Supplementary Table 1. The histogram of the the samples' age in different data collection cycles are shown in Supplementary Fig. 2.

In additioin, we use UK Biobank (https://www.ukbiobank.ac.uk/) samples as an external validation dataset. Ethics approval for the UK Biobank study was obtained from the North West - Haydock Research Ethics Committee (21/NW/0157). Informed consent was obtained from all UK Biobank participants (the consent form is available at https://www.ukbiobank.ac.uk/consent). For UK Biobank data, we include the 51 features that overlap (Supplementary Data 1) between the NHANES and UK Biobank datasets and have 384,762 samples with confirmed 5-year mortality status. All-cause mortality included deaths occurring before May, 2021. The dataset is randomly divided into training (80%) and testing (20%) sets. More detail about UK Biobank dataset is in Supplementary Methods and Supplementary Fig. 3.

**IMPACT framework.** To achieve high accuracy and explainable mortality prediction models, we developed the IMPACT (Fig. 1) framework, which combines tree-based models and TreeExplainer[24]. To model all-cause mortality, we use gradient boosted trees (GBTs). GBTs are nonparametric models composed of iteratively trained decision trees. The final ensemble of trees can capture non-linear and interaction effects between predictors. The hyperparameters are chosen by GridSearch and 5-fold cross-validation (Supplementary Methods). Model performance is measured using the area under the receiver operator characteristic curve (AUROC).

In our previous work, we introduced TreeExplainer[24], which provides a local (i.e., for each subject) explanation of the impact of input features on individual predictions for GBT models (Supplementary Methods). Specifically, TreeExplainer calculates exact SHAP[15] (SHapley Additive exPlanations) values for GBT models, which guarantee a set of desirable theoretical properties. SHAP values are additive; they sum to the model's output, i.e., the log-odds for GBTs. They are also consistent, which means

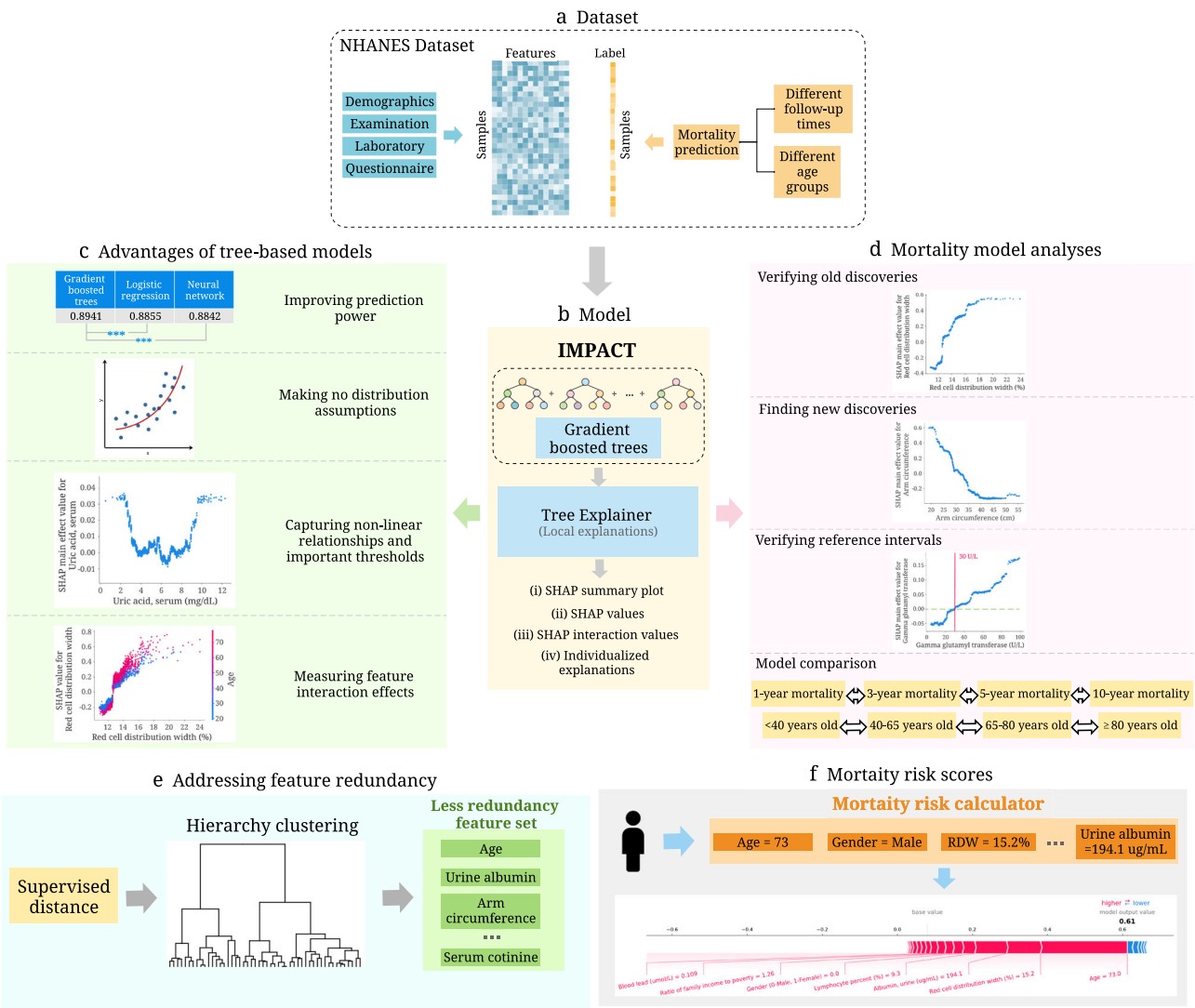

**Fig. 1 Overview of the IMPACT model and analyses. a** We use the NHANES (1999-2014) dataset, which includes 151 variables and 47,261 samples. The variables can be categorized into four groups: demographics, examination, laboratory and questionnaire. We train the model using different follow-up times and different age groups. **b** IMPACT combines tree-based models with an explainable AI method. Specifically, IMPACT (1) trains tree-based models for mortality prediction using the NHANES dataset, and (2) uses TreeExplainer to provide local explanations for our models. **c** We illustrate the advantages of interpretable tree-based models compared to traditional linear models in epidemiological studies. **d** We further analyze all mortality models and demonstrate the effectiveness of IMPACT at verifying existing findings, identifying new discoveries, verifying reference intervals, obtaining individualized explanations, and comparing models using different follow-up times and age groups. **e** We propose a supervised distance to help us explore feature redundancy. We further develop a supervised distances-based feature selection method that helps us select predictive and less-redundant features. **f** We build mortality risk scores that are applicable to professional and non-professional individuals with different cost-vs-accuracy tradeoffs. The individualized explanations of IMPACT show the impact of each risk factor for the overall risk score.

features that are unambiguously more important are guaranteed to have a higher SHAP value. Therefore, SHAP values are consistent and accurate calculations of each feature's contribution to the model's prediction. TreeExplainer also extends local explanations to capture pairwise feature interactions directly. In this work, we utilize TreeExplainer to conduct a systematic and integrated study of associations between a large number of variables and all-cause mortality. Here, higher SHAP values imply large contributions to mortality risk. By showing the impact of each variable and interactions among variables for local, sample-specific explanations, we can obtain a comprehensive understanding of why the model made a specific mortality prediction. The foreground samples and the SHAP values of the 1-, 3-, 5-, and 10-year mortality prediction models can be found in Supplementary Data 2–9.

In addition to studying the relationships between risk factors and all-cause mortality, we further propose a technique, "relative risk percentage", to identify sub-optimal reference intervals and a metric, "supervised distance", to measure feature redundancy and identify redundant groups of features given a specific prediction task. Building on supervised distance, we also propose a recursive feature selection strategy to select feature sets that are both predictive and less redundant. We additionally propose a recursive feature selection method to train accurate and efficient (low-cost) interpretable mortality risk scores.

**Supervised distance**
*Supervised distance and hierarchical clustering.* Supervised distance can accurately measure feature redundancy based on a specific prediction task. To calculate the supervised distance

**Table 1 Comparing the AUROCs between an existing mortality score or a biological age as reported in the original paper and the IMPACT-20 model tested for the corresponding follow-up time and age ranges in the NHANES dataset.**

| | Task | Age | AUROC | AUROC of IMPACT-20 | AUROC of IMPACT-20 (temporal validation) |
|---|---|---|---|---|---|
| **Mortality risk scores** | | | | | |
| Intermountain[6] | 1-year mortality | 18 + | 0.84 | 0.92 | 0.88 |
| Gagne Index[7] | 1-year mortality | 65 + | 0.79 | 0.85 | 0.85 |
| Intermountain[6] | 5-year mortality | 18 + | 0.87 | 0.89 | 0.88 |
| Prognostic score[5] | 5-year mortality | 40–70 | Male: 0.80 | Male: 0.85 | Male: 0.80 |
| | | | Female: 0.79 | Female: 0.83 | Female: 0.80 |
| Schonberg Index[8] | 5-year mortality | 65 + | 0.75 | 0.80 | 0.83 |
| **Biological ages** | | | | | |
| Horvath DNAm Age[17,19] | 10-year mortality | 21–84 | 0.56 | 0.90 | 0.89 |
| Hannum DNAm Age[18,19] | 10-year mortality | 21–84 | 0.57 | 0.90 | 0.89 |
| DNAm PhenoAge[19] | 10-year mortality | 21–84 | 0.62 | 0.90 | 0.89 |
| Phenotypic Age[19,67] | 10-year mortality | 20–85 | 0.88 | 0.90 | 0.89 |

The "AUROC" column shows the AUROCs reported in the original paper. The "AUROC of IMPACT-20" column shows the performance of IMPACT models trained with the selected top 20 features (Supplementary Tables 2 and 3). The "AUROC of IMPACT-20 (temporal validation)" column shows the performance of the IMPACT-20 models evaluated on the temporal validation set (Supplementary Methods).

between feature $i$ and feature $j$, we first train a uni-variate GBT model to predict the label (e.g. 5-year mortality in our study) using feature $i$. Then, we can obtain the $Prediction_i$ which is the output of the fitted uni-variate GBT. Next, we fit another uni-variate GBT to predict $Prediction_i$ using feature $j$. We define the output of the new GBT as $Prediction_i^j$. All hyperparameter values of the uni-variate GBTs are set to their default values. Following the same above steps, we can obtain $Prediction_j^i$. The supervised distance between feature $i$ and feature $j$ (supervised distance($i,j$)) is defined as:

$$supervised\ R^2(i,j) = max\left(0, 1 - mean\left(\frac{(Prediction_i - Prediction_i^j)^2}{var(Prediction_i)}\right)\right)$$
(1)

$$supervised\ distance(i,j) = max(1 - supervised\ R^2(i,j), 1 - supervised\ R^2(j,i))$$
(2)

where $var(x)$ is the variance of the vector $x$, $mean(x)$ is the average of the vector $x$. Supervised distance is scaled roughly between 0 and 1, where 0 distance means the features perfectly redundant and 1 means they are completely independent.

To explore the redundant feature groups, we hierarchically cluster all features according to the supervised distance. Specifically, we use complete linkage hierarchical clustering which merges in each step the two clusters whose merger has the smallest diameter.

*Supervised distance-based feature selection.* We propose a supervised distance-based feature selection method to select predictive and less-redundant feature sets. Firstly, we fit a GBT for 5-year mortality prediction on all features using the training set and rank the features by mean absolute SHAP values from TreeExplainer. We cluster features except age and gender into a specific number of groups using supervised distances-based hierarchical clustering and select the most important feature in each cluster. Then, we add age and gender to the selected feature set and re-fit the model. Next, we rerun the clustering using the new feature set except age and gender. This process is repeated until all remaining features cluster to a single group. In every iteration, we remove 5 features. The models are evaluated on the testing set with bootstrapping for 1000 times. We report the average of the AUROCs and the minimum supervised distance within the selected feature sets.

**5-year mortality risk scores.** IMPACT mortality risk scores are defined to be the prediction of the 5-year mortality prediction models. To compare with Intermountain gender-specific risk scores, we evaluate the models on different gender groups. The models are trained on the whole training set and evaluate on different gender groups in the testing set. Furthermore, considering the different feature collection cost for the general public and medical professionals, we build the risk scores starting from different feature sets. For the general public, the models are trained on all demographics, questionnaire features and examination features that are accessible at home for general public, For medical professionals, the models are trained on all demographics and laboratory features. We implement recursive feature selection to reduce the number of features included in the risk scores. Recursive feature elimination works by searching for a subset of features by starting with all features in the training dataset and successively removing features until the desired number of features remains. Firstly, we train a model on the full dataset with all features. Then we rank features by importance (mean absolute SHAP values) and remove the least important features. Another model is trained on the resulting feature set, and the process iterates until only the desired number of features are left. We remove 5 features in each iteration. We bootstrap the test set for 1000 times and assess the predictive performance. We report the average of the AUROCs within the selected feature sets.

**Reporting summary.** Further information on research design is available in the Nature Research Reporting Summary linked to this article.

## Results

**Advantages of tree-based models.** Linear models are commonly used in epidemiology because their coefficients indicate each feature's contribution to the model's prediction[25]. However, more expressive models, such as tree-based models, can achieve higher predictive accuracy across many datasets by learning non-linear relationships between features and the outcome variable. Gradient boosted trees (GBTs) have achieved state-of-the-art performance in many domains[26–29]. We observe the same trend in our study: tree-based models outperform both linear models and neural networks across almost all tasks we consider (Fig. 2a, Supplementary Fig. 4). The superior prediction performance of tree models indicates that we can capture signals relevant to mortality, which alternative approaches could not. Besides

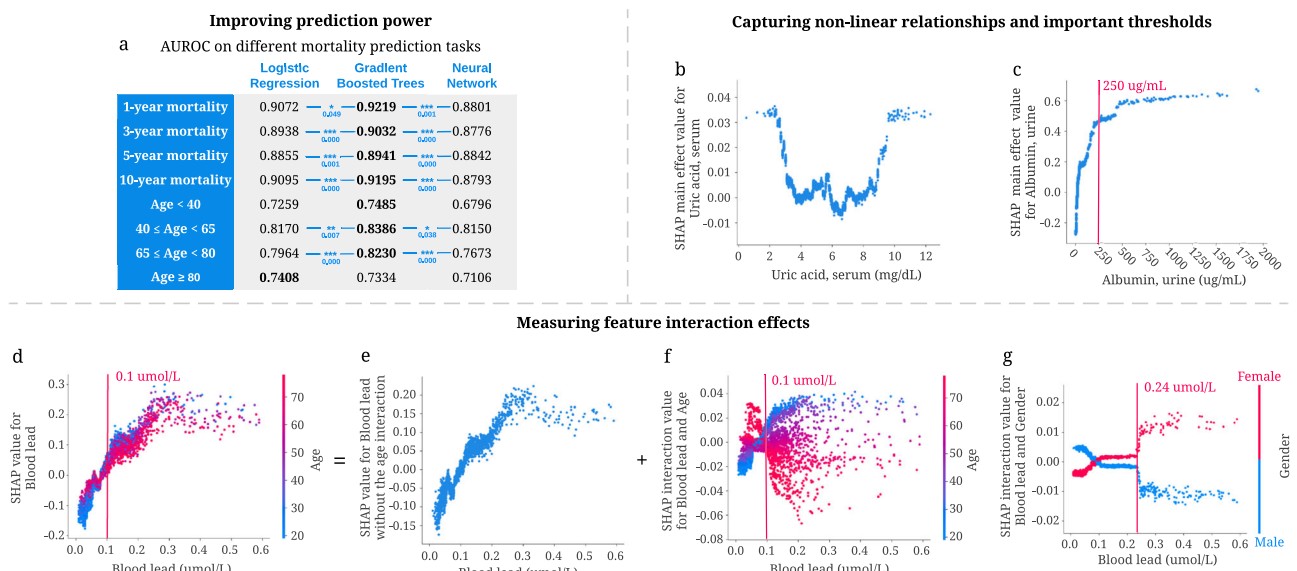

**Fig. 2 Advantages of tree-based models for mortality prediction. a** The area under the ROC curve (AUROC) of gradient boosted tree models outperforms both linear models and neural networks for seven of our prediction models. \*\*\**p*-value < 0.001, \*\**p*-value < 0.01, and \**p*-value < 0.05. *P*-values highlighted in blue are computed using bootstrap resampling over the tested time points while measuring the difference in area between the curves with $n = 1000$ independently resampling. **b**, **c** Tree-based models can capture non-linear relationships and important thresholds. **b** The main effect of uric acid on 5-year mortality. Higher SHAP value leads to higher mortality risk. **c** The main effect of urine albumin on 5-year mortality. **d**–**g** Tree-based models can measure feature interaction effects. **d** SHAP value for blood lead level in the 5-year mortality model. Each dot corresponds to an individual. The color corresponds to the value of a second feature (i.e., age) that has an interaction effect with blood lead. **e** We can use SHAP interaction values to remove the interaction effect of age from the model and obtain the SHAP value of blood lead without the age interaction on 5-year mortality. **f** Plotting just the interaction effect of blood lead with age shows how the effect of blood lead on mortality risk varies with age. **g** The SHAP interaction value of blood lead vs. gender in the 5-year mortality model.

predictive power, tree-based models have more advantages compared with traditional linear models. Our study illustrates the advantages of tree-based models in epidemiology, including making minimal assumptions, capturing non-linear relationships, important thresholds, and interaction effects.

*Tree-based models make minimal assumptions about the data distribution.* Several assumptions associated with linear models (e.g., linearity, independence, normality, etc.) constrain the features they can use. To satisfy these assumptions, scientists often manually transform non-linear variables before fitting a model (e.g., log-transformation, discretization of continuous variables, etc.). For instance, to explore the effect of blood lead on mortality, researchers first discretized blood lead using different thresholds. They observed that individuals with blood lead levels higher than the threshold had increased mortality risk compared to those with lower blood lead levels[30–32]. In comparison, tree-based models make minimal assumptions about the data distribution and need no data transformations. Figure 2d shows a positive relationship between blood lead and 5-year mortality risk. Tree-based models can capture complex relationships directly without needing to manually transform the variables.

*Tree-based models capture non-linear relationships and important thresholds.* Discovering non-linear relationships is important but challenging for epidemiological research using traditional linear models. J-shaped and U-shaped associations are two common and meaningful non-linear relationships[33]. However, linear models must use manually transformed features to capture non-linear relationships. As an example, Suliman et al.[34] used a linear model to show a J-shaped relationship between uric acid levels and mortality in patients with stage 5 chronic kidney disease (CKD) by dividing uric acid level into three categories and

calculating the hazard ratio for each. Unlike linear models, tree-based approaches can directly capture non-linear relationships. We observe a U-shaped relationship between uric acid level and all-cause 5-year mortality predictions in Fig. 2b. This relationship differs from the J-shaped one in previous work, possibly because of categorization, which loses essential information about values within the categories.

Additionally, discovering thresholds (i.e., inflection points beyond which changing a feature's value has diminishing returns) is important in epidemiological analysis. Figure 2c shows that $250\,\mu g/mL$ is an important threshold: according to our model, increasing urine albumin generally increases 5-year mortality risk; however, urine albumin higher than this threshold has almost the same impact on mortality risk.

*Tree-based models capture feature interaction effects.* Feature interaction examines how the effect of one feature on the outcome differs across strata of another feature, highlighting the complex relationship of two features on the outcome[35]. Tree-based models can naturally capture interaction effects by splitting on different features in the same tree. As shown in Fig. 2d–f, SHAP dependence plots can be decomposed into main effects and interaction effects for each sample. Figure 2f highlights a specific interaction: the relationship of blood lead level to mortality presents differently for young and old individuals. Specifically, for those with blood lead higher than $0.1\,\mu mol/L$, younger individuals have a higher 5-year mortality risk than older individuals. Figure 2g shows the SHAP interaction effects of gender with blood lead level: females have a higher 5-year mortality risk than males with blood lead levels higher than $0.24\,\mu mol/L$. The interaction effects of age and gender with blood lead level cannot be clearly identified without SHAP interaction values because being male or older generally increases mortality risk. These findings

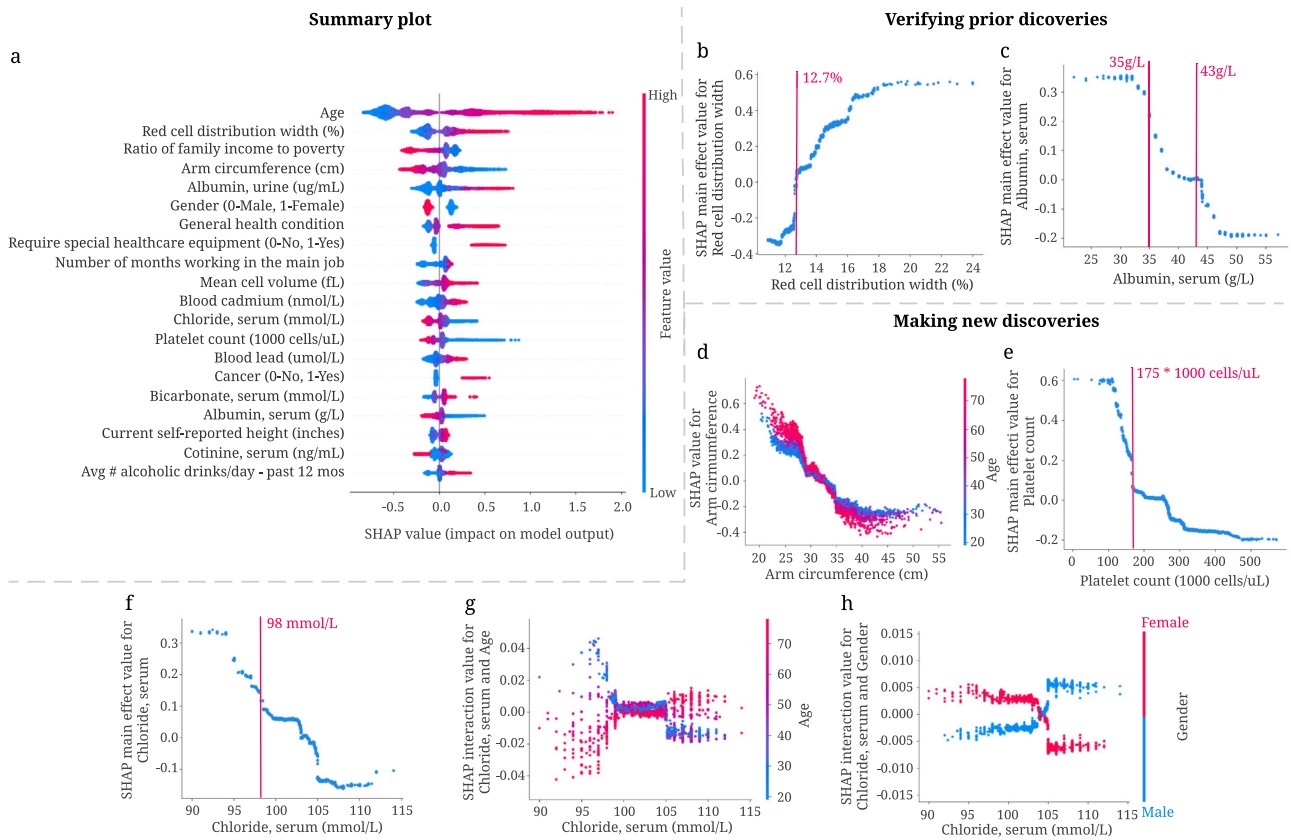

**Fig. 3 Combining 5-year mortality prediction gradient boosted trees models and local explanations to achieve significant discoveries about the entire model and individual features. a** SHAP summary plot for the gradient boosted trees trained on the 5-year mortality prediction task. The plot shows the most impactful features on prediction (ranked from most to least important) and the distribution of the impacts of each feature on model output, which includes a set of plots where each dot corresponds to an individual. The colors represent feature values for numeric features: red for larger values, and blue for smaller. The thickness of the line comprised of individual dots is determined by the number of examples at a given value. A negative SHAP value (extending to the left) indicates reduced mortality risk, while a positive one (extending to the right) indicates increased mortality risk. **b**, **c** IMPACT can verify well-studied features associated with mortality. **b** The main effect of red cell distribution width on 5-year mortality. **c** The main effect of serum albumin on 5-year mortality. **d**–**h** IMPACT can identify less well-studied features associated with mortality. **d** The SHAP value for arm circumference in 5-year mortality model. **e** The main effect of platelet count on 5-year mortality. **f** The main effect of serum chloride on 5-year mortality. **g** The SHAP interaction value of serum chloride vs. age in the 5-year mortality model. **h** The SHAP interaction value of serum chloride vs. gender in the 5-year mortality model.

underscore how being able to detect interaction effects can expose opportunities for further research.

**Discoveries from 5-year mortality prediction**. Figure 3a shows a summary plot that displays the magnitude, prevalence, and direction of the effect of the top 20 most impactful features on 5-year mortality prediction (Supplementary Methods). This summary plot provides an integrated explanation of the 5-year IMPACT model. Several features are known to be associated with mortality in epidemiological studies. Our results examine and support these studies' conclusions and surface additional discoveries, including features, thresholds, and non-linear relationships.

*IMPACT verifies well-studied features associated with mortality*. Some of the top 20 most important features for our 5-year mortality prediction models have been previously identified. For example, red cell distribution width (RDW), the second most important feature of the 5-year IMPACT model, has been shown to have a strong positive relationship with mortality in many studies under several conditions[36–39]. We also find a positive relationship between RDW and risk of mortality (Fig. 3b); moreover, 12.7% is an important threshold over which RDW

manifests a positive effect on mortality. Serum albumin level's relation to mortality is also well-studied; previous studies show that serum albumin is negatively associated with mortality risk[40–42]. The relationship shown in Fig. 3c matches this trend. Furthermore, Corti et al.[40] showed that serum albumin level < 35 g/L was associated with an increased risk of mortality compared to serum albumin levels greater than 43 g/L[40]. We observe that 35 g/L and 43 g/L are indeed key inflection points (Fig. 3c): serum albumin levels lower than 43 g/L have a positive relationship with mortality prediction, while those around 35 g/L are associated with a dramatically increased mortality risk.

*IMPACT identifies less well-studied features associated with mortality*. Some of the top 20 most important features identified by IMPACT are less appreciated as mortality risk factors in the existing epidemiological literature. Three of these are arm circumference, platelet count, and serum chloride level. Figure 3d shows a negative relationship between arm circumference and 5-year mortality, especially for older people. This negative relationship is consistent with previous work[43,44]. IMPACT ranks arm circumference as the fourth most important feature for 5-year mortality prediction, with an importance ranking that greatly exceeds that of BMI (the 56th). This suggests that smaller

**Table 2 Providing additional perspective to laboratory reference intervals.**

| Feature | Reference Interval | Relative Risk Percentage (RRP) | | | |
|---|---|---|---|---|---|
| | | 1-year | 3-year | 5-year | 10-year |
| Gamma glutamyl transferase | 0–30 U/L | 16.93% | −4.57% | −0.97% | −6.04% |
| Globulin, serum | 20–35 g/L | 5.39% | 7.95% | 14.73% | 4.59% |
| Lymphocyte percent | 20%–40% | 15.63% | 7.02% | 6.55% | 10.81% |
| Blood urea nitrogen (Male) | 2.86–8.57 mmol/L | 8.12% | 2.92% | 8.02% | 21.08% |
| Blood urea nitrogen (Female) | 2.14–7.50 mmol/L | −0.15% | 3.07% | 0.40% | 12.16% |
| Albumin, serum | 35–50 g/L | 28.56% | 49.70% | 59.77% | 93.48% |
| Blood lead | 0–0.48 umol/L | 100.00% | 94.71% | 100.00% | 100.00% |
| Mean cell volume | 80–100 fL | 82.80% | 75.82% | 83.92% | 57.26% |
| Alanine aminotransferase ALT (Male) | 7–55 IU/L | 100.00% | 100.00% | 100.00% | 100.00% |
| Alanine aminotransferase ALT (Female) | 7–45 IU/L | 100.00% | 100.00% | 100.00% | 100.00% |

The table lists the reference interval and relative risk percentage (RRP) of the selected laboratory features. RRP measures the relative risk of the feature values within the reference interval compared to the relative risk of all values. A higher RRP indicates that the current reference interval is relatively more inappropriate. The negative value indicates that the reference interval of that laboratory feature is optimal for mortality risk. The 100% value suggests that the reference interval may be sub-optimal for mortality risk.

arm circumference is more predictive than BMI for modeling mortality, as in[45].

Figure 3e shows a negative relationship between platelet count, the 13th most important feature, and 5-year mortality. $175 \times 1000$ cells/$\mu$L is an important threshold; platelet count lower than that level is associated with dramatically increased mortality risk. Serum chloride is also inversely related to 5-year mortality (Fig. 3f). The normal adult value for chloride is 98-106 mmol/L. We observe that serum chloride lower than 98 mmol/L is associated with sharply increased mortality risk. In Fig. 3g–h, we plot the interaction effect of age and sex with serum chloride level. This analysis reveals that younger people and females with low serum chloride have a higher mortality risk than older people and males. The interaction effect of age and serum chloride shows that early rather than late-onset low chloride level has a greater effect on the model.

*IMPACT can provide an additional perspective to laboratory reference intervals.* A reference interval (RI) is the range of values that is deemed normal for a physiologic measurement in healthy persons[46]. It is the most commonly used decision support tool to interpret patient laboratory test results. RIs enable differentiation of healthy and unhealthy individuals[47,48]. Hence, the quality of the RIs is as crucial as the quality of the result itself. RIs in use today are most commonly defined as the central 95% of laboratory test results in a reference population. Unfortunately, this definition does not consider mortality or disease risk, which may lead to misdiagnosis since RIs are often used to identify unhealthy individuals. The partial dependence plots (Supplementary Methods) of IMPACT models directly reflect the effects of the features on mortality risk, which provides an alternative perspective for identifying inappropriate reference intervals with mortality/disease relevance.

We define the relative risk percentage (RRP; Supplementary Methods) that measures the relative risk of the feature values within the reference interval compared to the relative risk of all values (Table 2). A higher RRP indicates that the feature values within the reference interval may lead to high mortality risk, which call for special attention. The first four features in Table 2 have relatively low 5-year mortality RRP. From Fig. 4a–e, we observe that the values of these features within the reference interval have a low 5-year relative mortality risk; the values outside the reference interval may lead to increased 5-year mortality risk. Therefore, IMPACT confirms the reference intervals of these four features as optimal for mortality risk. In contrast, the RRPs of the last four features in Table 2 are high. Figure 4f–j also shows that the relative 5-year mortality risk of the

values within the reference interval is high compared to the maximum relative risk of all values. Hence, IMPACT identified the divergence where reference intervals appear to be poorly tuned to mortality risk, suggesting that these reference intervals may in fact be sub-optimal for health. Note that our goal is not to suggest the optimal reference range: to find the optimal reference interval, more careful sample and study design need to occur. The partial dependence plots for the 1-, 3- and 10-year mortality prediction models are shown in Supplementary Fig. 5.

*External Validation of IMPACT on UK Biobank (UKB) dataset.* We validate the key findings of the 5-year mortality prediction IMPACT model using the UKB dataset. Our external validation includes two aspects. The first aims to validate the entire IMPACT framework using a new dataset by checking whether the explanations from a model trained on the NHANES dataset can also be found in a model trained on the UKB dataset. The second aims to test the generalizability of the mortality prediction model trained on the NHANES dataset.

To validate the IMPACT framework, we train a tree-based 5-year mortality prediction model on the UKB dataset using the 51 overlapping features between NHANES and UKB. Then, we calculate the SHAP values of the UKB mortality prediction model using the UKB samples. Figure 5a shows the relative global feature importances of the 51 overlapping features of the NHANES model (trained on all 151 features) and the UKB model (trained on 51 features). We can see that the top 20 most important features are largely consistent, where 14 features are the same for both models. The *p*-value of the Fisher's exact test ($p = 0.0004$) shows that the overlap between the top 20 most important features of NHANES (151 features) and UKB (51 features) model is significant. The Spearman's correlation coefficient of the NHANES and UKB model's feature importance is 0.6654 (*p*-value < 0.0001), showing the significant positive correlation between the ranking of the overlapping features in NHANES and UKB. It is worth mentioning that waist circumference is more important than BMI in the UKB model, which further validates that some anthropometric measures (i.e., arm circumference in the NHANES model, waist circumference in the UKB model) are more predictive than BMI for modeling mortality. Figure 5b–d show the relationship between 5-year mortality and three important features: red cell distribution width, serum albumin, and serum uric acid. The trends discovered by the SHAP main effects in the UKB model corroborate previous findings from the NHANES model. In Fig. 5e, f, the values of gamma glutamyl transferase and lymphocyte percent in the reference interval have a low 5-year

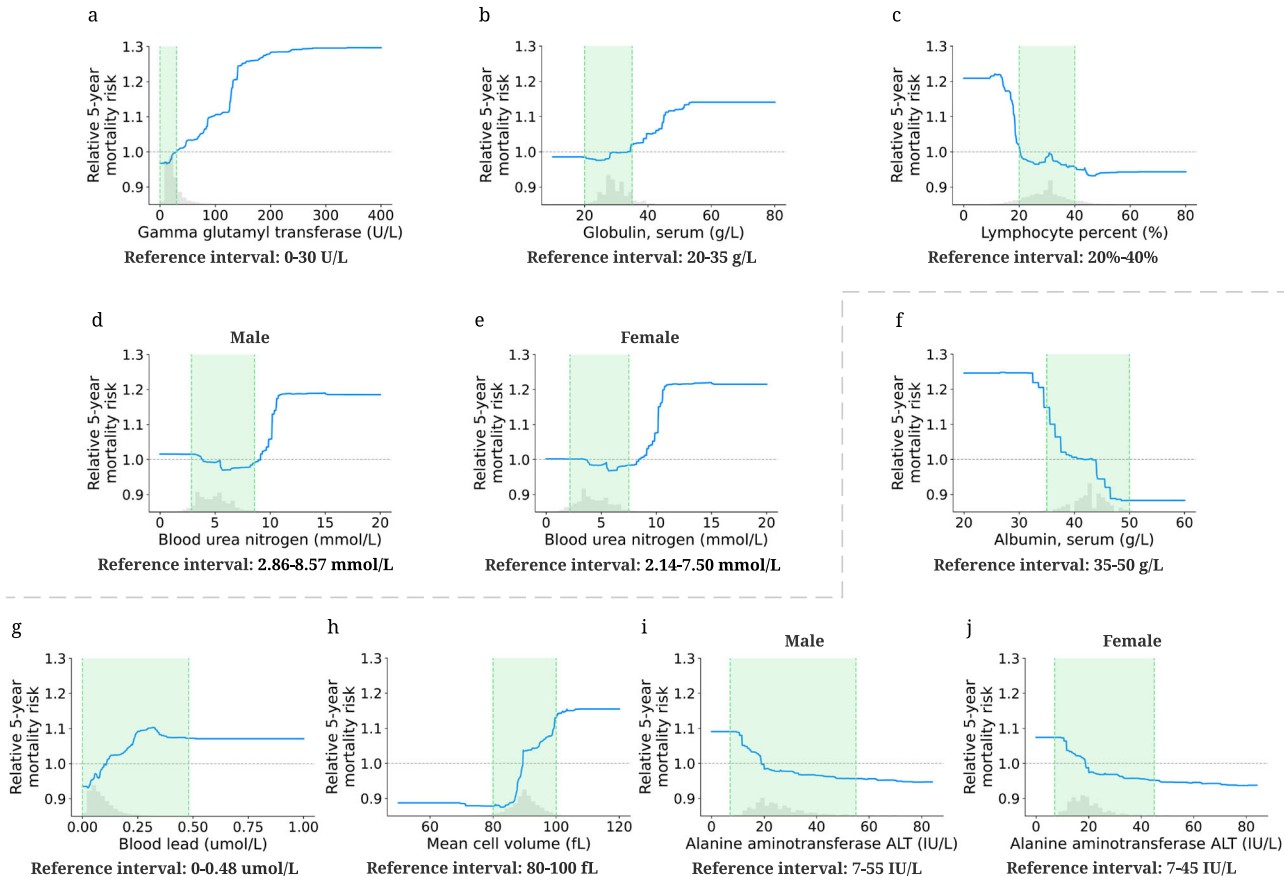

**Fig. 4 Effect of varying laboratory feature values on 5-year mortality risk.** The partial dependence plots show the change in relative 5-year mortality risk for all values of a given laboratory feature. The grey histograms on each plot show the distribution of values for that feature in the test set. The green shaded region shows the reference interval of each feature. The grey dotted line shows the average value of the model predicted probability ($y = 1$).
**a–e** The partial dependence plots of the features whose reference intervals are optimal for mortality risk. **f–j** The partial dependence plots of the features whose reference intervals are sub-optimal for mortality risk.

relative mortality risk, which demonstrates that the reference intervals of these two features are optimal for mortality risk. In contrast, Fig. 5g shows that the relative 5-year mortality risk of the values of serum albumin in the reference interval is high, which suggests that the reference interval may be suboptimal for health. These results are consistent with our findings from the NHANES model trained on 151 features. More validation results of IMPACT framework on the UKB dataset are in Supplementary Fig. 6.

Furthermore, we would like to validate whether the performance and explanations of the NHANES prediction model generalize to an unseen population (UKB). Training details and results are described in Supplementary Note 1, Supplementary Figs. 7, 8. Our external validation results show that the NHANES mortality prediction model generalizes well to the UKB dataset in terms of both mortality prediction performance and key relationships between features and mortality.

**Discoveries for mortality prediction using different follow-up times.** The relationship between each feature and mortality may change for different models. For instance, comparing important features between IMPACT models using different follow-up times can reveal features that are predictive only for short-term mortality, not longer-term mortality (and vice versa).

Figure 6a shows the top 20 most important features and relative importance of input features in IMPACT's 1-year, 3-year, 5-year, and 10-year mortality prediction models. Feature importance rankings change greatly between these four models.

Some features are important for all four (e.g., age, RDW, and urine albumin level). Some features become more important over time (e.g., platelet count, whose importance ranking is 75 for the 1-year model and 12 for the 10-year model). Other features become less important over time (e.g., serum potassium, whose importance ranking is 17 for the 1-year model and 42 for the 10-year model). These results provide a more comprehensive understanding of shorter- and longer-term mortality risk, which can facilitate the investigation of mechanisms underlying risk predictors and potentially help validate interventions.

The relationship between each feature and mortality may change for models that predict different mortality outcomes or utilize different subsamples of the general population. For instance, Fig. 6b, c show the SHAP value for serum potassium in IMPACT's 1-year and 5-year mortality prediction models. The finding that serum potassium lower than 3.5 mmol/L and higher than 4.0 mmol/L are associated with increased mortality risk has been previously observed[49–51]. Interestingly, for the 1-year model, hyperkalemia (high potassium) has a higher mortality risk than hypokalemia (low potassium). For the 5-year model, hypokalemia has the same or higher mortality risk than hyperkalemia. Figure 6d shows that serum sodium higher than 139 mmol/L increases 1-year mortality risk, and low serum sodium with negative SHAP values decreases mortality risk. However, the relationship differs completely in the 5-year mortality prediction model (Fig. 6e): hyponatremia (serum sodium < 135 mmol/L) is associated with a higher 5-year mortality risk. This type of insight,

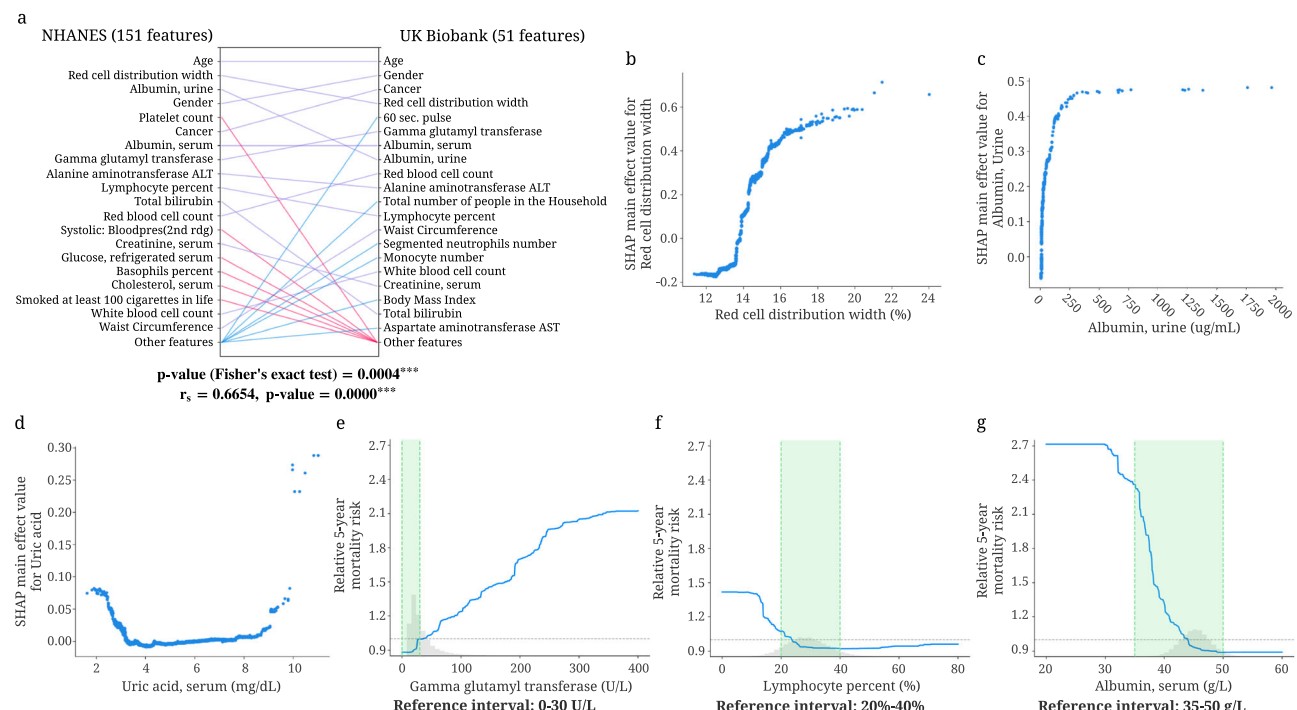

**Fig. 5 External validation of IMPACT framework on the UKB dataset. a** Relative importance of 51 overlapping features in the 5-year mortality prediction models trained on the NHANES (151 features) and UKB (51 features) datasets. For each model, the figure shows the 20 most important features of prediction (ordered by importance). The purple line indicates that the feature is in the top 20 features of both models. Blue and red lines indicate the feature is in the top 20 features of one model but not the other. The *p*-value of the Fisher's exact test examines the overlap between the top 20 most important overlapping features in the NHANES and UKB models (the contingency table in Supplementary Figure 6F). The Spearman's correlation coefficient is calculated using the feature importance of the overlapping features in NHANES and UKB (*n* = 51 featurs). ***p*-value < 0.001. **b–d** The main effect of red cell distribution width, urine albumin and serum uric acid on 5-year mortality in the model trained on UKB (51 features) dataset. **e–g** The relative 5-year mortality risk of gamma glutamyl transferase, lymphocyte percent, and serum albumin in the model trained on the UKB (51 features) dataset.

especially regarding the differences of non-linear trends, is not apparent using linear models.

Likewise, we can compare models trained on distinct subpopulations (e.g., samples in different age groups). The differences between these models can help researchers identify risk predictors relevant to each subpopulation and provide epidemiological insights that may guide policy for specific at-risk populations. The discoveries for mortality prediction using different age groups are discussed in Supplementary Note 2 and Supplementary Fig. 9. We further explore explaining the mortality predictions using different age distributions in Supplementary Note 3, Supplementary Fig. 10.

**Exploring feature redundancy using supervised distance**. Features in datasets are often partially or fully redundant with each other, such that a model could use either feature and achieve the same accuracy. It is important to be aware of redundant features when we interpret a model because these features may include the same information about the output and thereby split the importance of this information. To this end, we propose a supervised distance, which helps us explore and better understand redundant features (Supplementary Methods). Building upon supervised distance, we develop a feature selection method to maximize accuracy and minimize redundancy.

*Supervised distances measures feature redundancy and identifies redundant groups of features.* Researchers often use unsupervised

methods, such as some form of correlation-based clustering, to identify dependent features[52,53]. However, when we have a specific prediction task in mind, we would like to measure feature redundancy with respect to outcome. This can be done using supervised distance, which measures the similarity of two features' information about the prediction task by training one univariate model to predict the outcome of another (Supplementary Fig. 11; Supplementary Methods). Supervised distance is scaled roughly between 0 and 1, where 0 distance means the features are perfectly redundant regarding the prediction task and 1 means they are not redundant at all.

To identify groups of redundant features, we hierarchically cluster all features according to supervised distance (Supplementary Fig. 12; Supplementary Methods). Redundant features with the same information about the output group together. For example, arm circumference, the fourth most important feature of the 5-year IMPACT model, is grouped with weight-related features: BMI, waist circumference, weight, etc. These weight-related features all contain similar information about 5-year mortality. To further explore the predictive ability of the features, we train models using one weight-related feature and all non-weight-related features (reducing redundancy models) and models using one weight-related feature in addition to age and gender (single-feature models) (Supplementary Methods). Arm circumference is the most predictive weight-related feature across all settings (Fig. 7a), and may be more informative than other weight-related features with respect to all-cause mortality. Another example is the cluster that includes many blood test

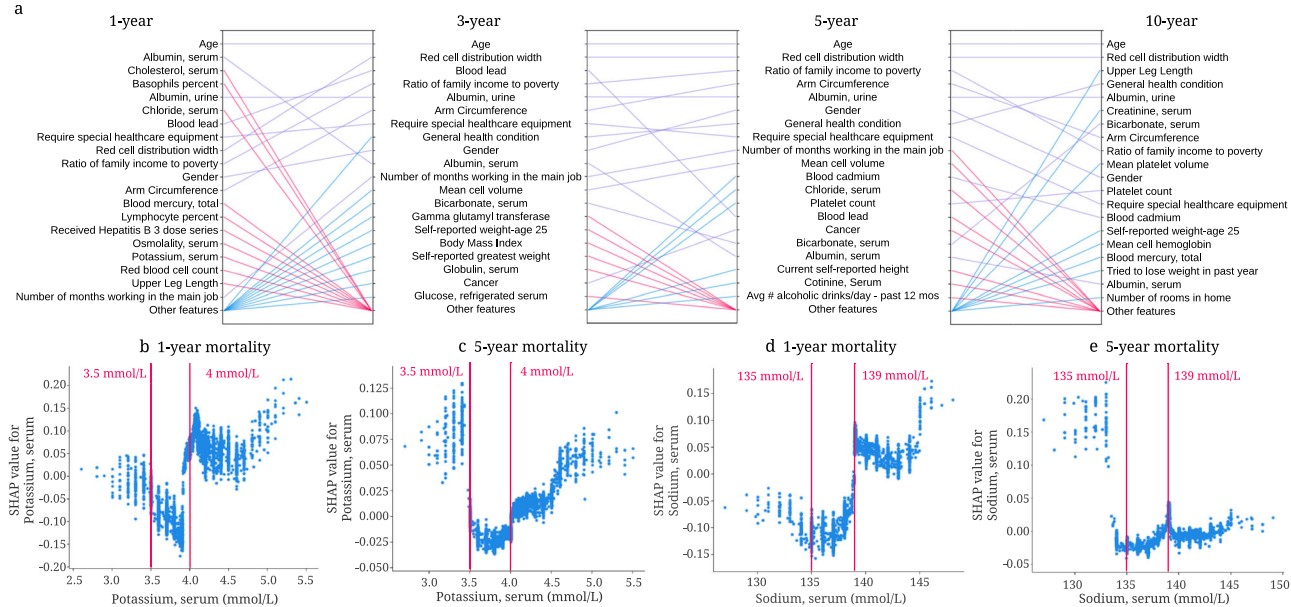

**Fig. 6 Understanding important risk factors for mortality prediction from tree-based models based on different follow-up times. a** Relative importance of input features in 1-, 3-, 5- and 10-year mortality models. For each model, the figure shows the 20 most important features of prediction (ordered by importance). The purple line indicates that the feature is in the top 20 features of two models. Blue and red lines indicate that the feature is in the top 20 features of one model, but not in the top 20 features of the other. **b** The SHAP value of serum potassium in the 1-year mortality model. **c** The SHAP value of serum potassium in the 5-year mortality model. **d** The SHAP value of serum sodium in the 1-year mortality model. **e** The SHAP value of serum sodium in the 5-year mortality model.

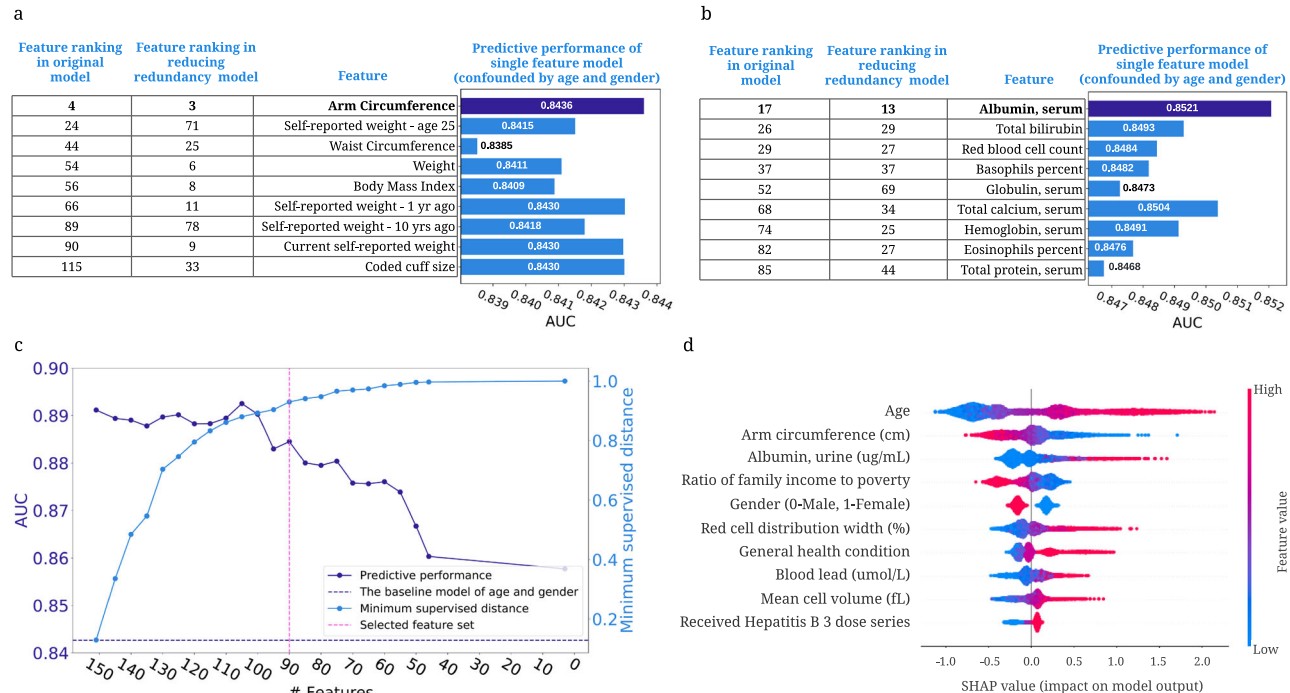

**Fig. 7 Exploring feature redundancy using supervised distance. a** The feature importance ranking of the BMI-related features in original models and reducing redundancy models (models using one weight-related feature and all non-weight-related features), and the AUC of the single-feature models controlling for age and gender. **b** The feature importance ranking of the selected laboratory features in original models and reducing redundancy models, and the AUROC of the single-feature models confounded by age and gender. **c** The AUROC of the models using the selected feature sets and minimum feature redundancy within the selected feature sets when running supervised distance-based feature selection. The purple dashed line shows the AUROC of the model trained on age and gender. The pink dashed line indicates the feature set we select for further analysis. **d** The SHAP summary plot for the gradient boosted trees trained on the selected 90 features for the 5-year mortality prediction.

features (Fig. 7b). Similar to arm circumference, serum albumin is the most predictive feature among these blood test features. In summary, using supervised distance, we can easily identify redundant feature groups and select the most representative feature based on predictive power. These selected features can be the strongest risk predictors because they have strong predictive power and can represent a number of features.

*Supervised distance-based less-redundant feature selection.* To address feature redundancy more rigorously, we propose a recursive feature selection method to select predictive and less redundant feature sets based on supervised distance (Supplementary Methods; Supplementary Data 1). Figure 7c shows the predictive power and minimum supervised distance of subsets of features refined by our feature selection approach. We observe that as the number of features declines, the predictive performance drops, and the feature redundancy reduces (as indicated by an increasing minimum supervised distance). The figure shows that when using 90 features, the model can achieve good predictive performance (AUROC = 0.8845), and the minimum supervised distance within the features is high (0.9301). Figure 7d shows the summary plot of the top 10 features in the 5-year mortality prediction model using the selected 90 features. Since there is less redundancy in the selected features, we mitigate the issue of redundant features splitting credit. This lets us explore more richly the effect of important risk predictors on mortality. In our low redundancy model, arm circumference is selected to represent the weight-related features and still receives high importance. Furthermore, we find that "requiring special healthcare equipment," a top 10 feature in the model trained on all features, is removed from the feature list because it is redundant with "general health condition." In summary, our feature selection method helps remove redundant features while retaining highly predictive features, thereby balancing accuracy and interpretability.

**Highly accurate and efficient interpretable mortality risk scores.** A mortality risk score can help individuals monitor their health status, clinicians stratify high-risk patients, and public health organizations guide policy. Most prior mortality risk scores are built with linear models, such as logistic regression and linear hazard models[5,6]. However, compared with traditional models, tree-based models achieve higher predictive performance, which can stratify patients better than linear models (Table 1). Besides predictive performance, we must also consider the feature collection cost. There is a tradeoff between collecting fewer features (which is less costly) and model performance (cost-vs-accuracy). Moreover, the cost of features differs for different users. For example, blood test features are easily collected by clinicians, but, for the public, questionnaire features and examination features are easy to obtain at home. Furthermore, in addition to calculating their risk scores, users may want to know which features contributed more or less to their risk. To address these problems, we build interpretable tree-based mortality risk scores with different cost-vs-accuracy tradeoffs and different types of features for the general public (demographic, examination, and questionnaire features) and medical professionals (demographic, laboratory features and features from common test panels) to use (Supplementary Methods; Supplementary Data 1). Compared with previous mortality risk scores, ours are more interpretable, more accurate, applicable to more users, and flexible with respect to different cost-vs-accuracy tradeoffs.

*IMPACT develops highly accurate and efficient 5-year mortality risk scores.* The predicted probability of IMPACT models can be directly used as mortality risk scores (IMPACT risk scores). We did a temporal validation of the risk scores by training and validating them in samples from NHANES 1999–2008 and assessing their performances in NHANES 2009–2014. The sample size, the number of deceased samples and the histogram of age in the training set with the testing set and the temporal validation set are shown in Supplementary Fig. 13. For comparison, we train linear and tree-based Cox proportional hazard models widely used in previous work (Supplementary Methods). To find less costly but nearly as accurate models, we select the features using recursive feature elimination (RFE; Supplementary Methods). Moreover, we compare IMPACT risk scores with Intermountain sex-specific risk scores[6] (Supplementary Methods). The models are evaluated on different gender groups.

In Fig. 8a, b, we show the AUROC of the 5-year mortality risk scores of female samples (Supplementary Fig. 14 for male results) in the test set and the temporal validation set. We see that the IMPACT model with only 20 features obtains an AUROC of 0.8971, which is almost as same as the performance of the model using all features (AUROC = 0.9030); using fewer than 20 features leads to a dramatic accuracy drop. Figure 8a, b also show that IMPACT models achieve better performance than linear and tree-based Cox proportional hazard models. Furthermore, we see that the IMPACT risk score using the laboratory features (AUROC = 0.8881) and the risk score using the questionnaire and examination features (AUROC = 0.8835) both achieve acceptable predictive performance. The IMPACT risk score using the features from common test panels achieves higher AUROCs than the intermountain risk score, which uses CBC and BMP panels features. With the models trained with different cost-vs-accuracy tradeoffs, users who cannot measure certain features (i.e., high-cost features) can still calculate accurate mortality risk scores. Figure 8b shows that the performance of our models drops only a little on the temporal validation set, which can indicate that our risk scores generalize fairly well. The selected top 20 features and features from CBC and BMP panels are listed in Supplementary Table 2. In summary, we build IMPACT risk scores that are applicable to professional and non-professional individuals with different cost-vs-accuracy tradeoffs.

*IMPACT exposes individualized mortality risk score explanations.* TreeExplainer can help researchers analyze the prediction for each individual and illustrate each features' contribution to the mortality risk score. We explain the mortality prediction model in terms of its probability predictions (risk scores). Figure 8c, d show individualized explanations for two people in the model using the top 20 features (Supplementary Methods). The first individual (Fig. 8c) was alive after 5 years. From the figure, we observe that IMPACT predicted that the individual's 5-year mortality risk score was 0.02, lower than the average predicted risk (i.e., base value). Certain features can increase mortality risk, such as red cell distribution width, and others can decrease it, such as urine albumin level. For this individual, the features that drive down mortality risk outweigh those that increase it. The second individual (Fig. 8d) was deceased after 5 years, and the model's predicted mortality probability is 0.61, much higher than the average predicted risk. The top three features that increase this individual's risk are high age, high red cell distribution width, and high urine albumin concentration. The interpretable risk score can both help individuals improve health awareness and understand their health status, and it can help health professionals identify high-risk individuals.

## Discussion
IMPACT combines high-accuracy complex ML models and state-of-the-art local explanation methods to allow the systematic study

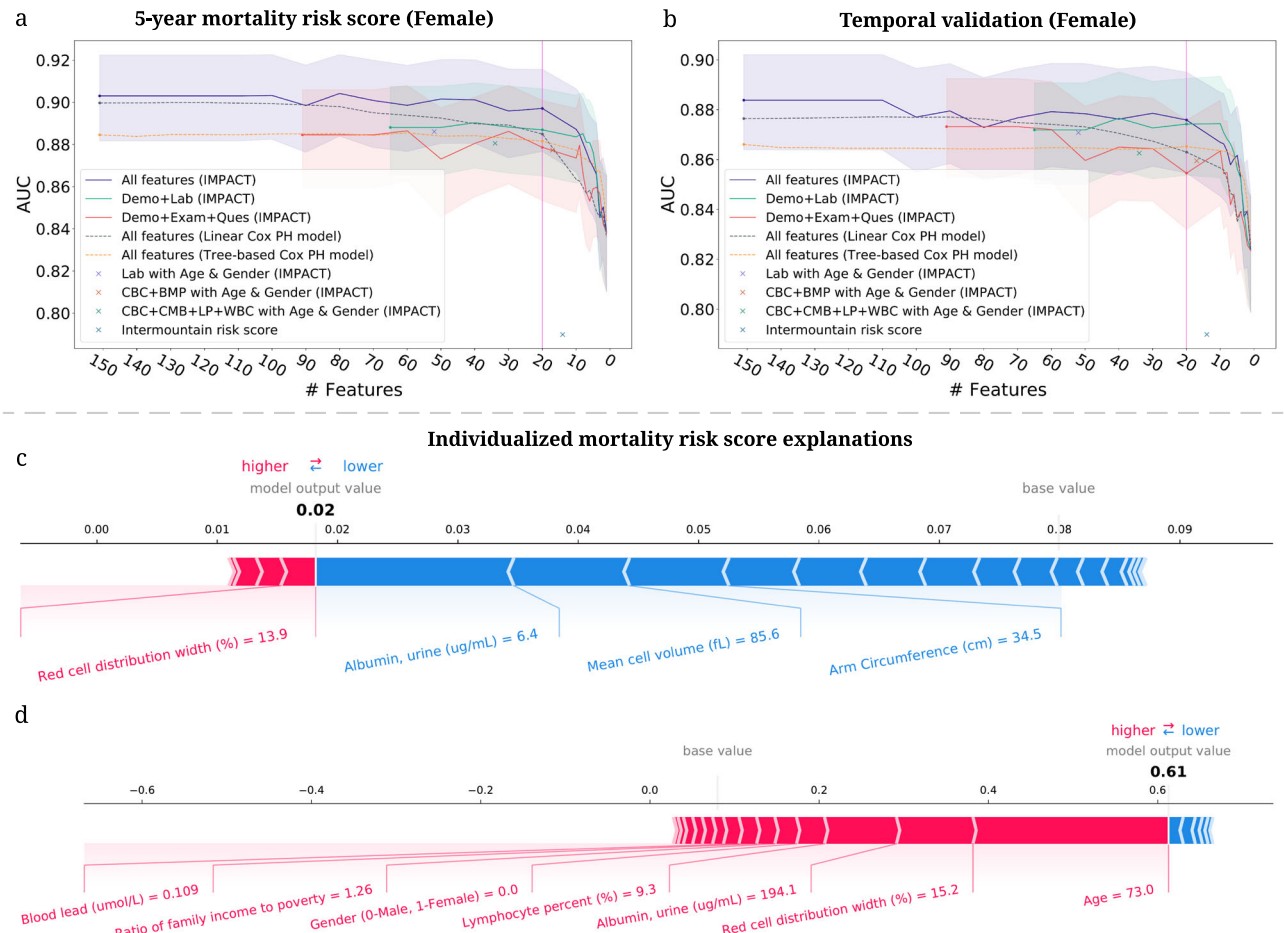

**Fig. 8 Developing highly accurate and efficient interpretable 5-year mortality risk scores. a, b** The AUROC of the models using different feature sets after recursive feature elimination. Lines are mean performance over 1000 random train/test splits, and shaded bands are 95 percent normal confidence intervals. **a** The AUROC of the models tested on the female group in the test set of NHANES 1999–2008. **b** The AUROC of the models tested on the female group in the temporal validation set (NHANES 2009–2014). **c, d** IMPACT can analyze individualized mortality risk scores. **c** The individualized explanation for an individual who is alive after 5 years. The output value is the risk score for that individual. The base value is the mean risk score, i.e., the score that would be predicted if we did not know any features for the current output. The features in red increase mortality risk, and those in blue decrease it. **d** The individualized explanation for a sample who is deceased after 5 years.

of all-cause mortality. In epidemiology, high accuracy is neccessary but insufficient; explaining models to humans is also essential for drawing epidemiological hypotheses[54,55]. IMPACT's combination of accuracy and explanation aims to optimize accuracy while also gaining insight into complex interrelations between mortality and an individual's features.

Using 151 features in NHANES 1999–2014, we build tree-based mortality prediction models and explore the effect of those features on mortality for different follow-up times and age groups. Importantly, we demonstrate the value and significance of explaining complex ML prognostic models. IMPACT lets us to capture both non-linear and interaction effects that are difficult to uncover with linear models. These results help us verify well-studied findings (e.g., the relationship of red cell distribution width and serum albumin with mortality) as well as identify less well-studied ones (e.g. the important risk predictors arm circumference, platelet count and serum chloride, and the complex interactions among the features). One pitfall to inferring relationships between determinants and an outcome are relationships between the determinants themselves (redundancy). To address this, we propose a supervised distance and feature selection approach, which we utilize to select the minimally redundant feature sets. Finally, we build easy-to-use and explainable

mortality risk scores for use by both the general public and medical professionals with different tradeoffs between feature collection cost and model performance. These scores can help individuals improve self-awareness of their health status and help clinicians identify patients with high mortality risk to target with specific interventions. In this paper, we present only a small part of our findings. All our results and risk scores are available for public use in an interactive website (https://suinleelab.github.io/IMPACT), where associations and interactions can be explored in detail to generate new research hypotheses.

In terms of epidemiological findings, this study shows a negative relationship between arm circumference and mortality. Our clustering method groups arm circumference with BMI and other weight-related features, indicating that these features share information about mortality. Several prior studies have found a U-shaped association between BMI and mortality, where very low or very high BMI is associated with greater mortality risk[43,56]. This U-shaped relationship may be the result of compound effects from body fat and fat-free mass. Since upper arm circumference is an indicator of fat-free mass[43,44], it may be the case that fat-free mass is driving the inverse correlation between arm circumference and mortality risk. Larger arm circumference is expected to be associated with greater muscle mass, while smaller

arm circumference may reflect muscle deterioration along with diminished nutritional status or malnutrition[57,58]. The importance of arm circumference in IMPACT is consistent with previous studies, which show that low arm circumference was more effective than low BMI in predicting follow-up mortality risk in older people[57,59,60].

One limitation of IMPACT is that the relationships and interactions detected by our model cannot be claimed to be causal. This is not unique to our method and poses a fundamental problem in epidemiological studies using observational data. The purpose of this study is not to address causality, but rather to conduct a systematic study of mortality associations with the NHANES population. In particular, a primary obstacle in capturing causal effects with observational data and predictive models are confounding variables. In order to condition on confounders (and potential surrogate confounders), it is often desirable to include as many features as possible in the model[61]. Conversely, we may want to remove colliders and mediators that skew the real effect of treatment features of interest. Our solution to redundancy, i.e., supervised distance, can potentially help narrow down related features for which domain experts can identify colliders, mediators, and confounders. This is a potential future research question that takes a step in the direction of making explanations from complex models causal.

Our study is performed on NHANES 1999–2014 data, which is designed to assess the health status of participants in the United States. We perform temporal validation within the NHANES samples to evaluate the performance of our mortality risk scores. To evaluate the generalizability of important features and relationships, we implement the IMPACT model on a geographically distinct dataset with samples exclusively from the United Kingdom (UK Biobank). Although our qualitative findings were consistent between NHANES and UK Biobank, there are differences between both populations, primarily in terms of age (37–72 in UKB vs. 18–80+ in NHANES), which also affects the base rates of mortality in each data set. As such, further external validation of our mortality models on datasets with similar distribution of variables and mortality rates should be undertaken to further increase the generalizability of the findings.

Over the past several years, a variety of ML approaches have been applied in the field of aging research to develop "clocks" that can predict the chronological age of an individual based on different phenotypic features[62]. The most common of these are the epigenetic clocks that have identified patterns of methylation on DNA that change with age and can be used to predict chronological age with high accuracy across a variety of different species and tissue types[63,64]. Other clocks based on gene expression, metabolites, facial features, telomere length, etc., have also been described[65]. Efforts have also been made to use these clocks to predict an individual's biological age, which may differ from their chronological age if they are aging more rapidly or slowly than the general population. Such "biological aging clocks" are expected to reflect the underlying health status of the individual and be useful for predicting future health outcomes and mortality. Although we have not yet attempted to validate IMPACT as a tool for assessing biological age, those individuals with lower IMPACT mortality risk than expected for their chronological age would be predicted to have a lower biological age, and vice-versa. Because IMPACT is trained to predict all-cause mortality rather than fit to chronological age, it will be of interest to determine how IMPACT compares to these various clocks in predictive capacity, particularly if done for the same cohort of individuals.

Prognosis research using complex ML models will likely increase over the coming years as ML techniques continue to rapidly develop. However, "black box" ML models that predict without explaining are difficult for clinicians to trust and difficult to extract meaningful information from. Therefore, the combination of complex ML models and 'explainable artificial intelligence' (XAI) is necessary and urgent. IMPACT takes a consequential step towards XAI for mortality prediction. This study's improvement in predictive accuracy and explanation of complex ML models warrants further exploration for other epidemiological outcomes.

## Data availability

No new data are generated in this study. The NHANES survey is a public-use data files prepared and disseminated to provide access to the full scope of the data. The NHANES data for all experiments in the paper is publicly available at https://www.cdc.gov/nchs/nhanes/index.htm. A downloadable version of the dataset is available at https://github.com/suinleelab/IMPACT. The UK Biobank data used in this study is obtained via material transfer agreement as part of Data Access Application 59898. All data is available by UK Biobank via their standard data access procedure (http://www.ukbiobank.ac.uk/register-apply). The data underlying all figures are available in Supplementary Data 2–9.

## Code availability

All code for our study, including code to train the mortality prediction models and to generate all figures included in the manuscript, are available at https://github.com/suinleelab/IMPACT (archived at https://doi.org/10.5281/zenodo.6899541[66]).

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

## Acknowledgements

This work was funded by National Science Foundation [DBI-1759487, DBI-1552309, DBI-1355899, DGE-1762114]; National Institutes of Health [R35 GM 128638, R01 NIA AG 061132 and P30 AG 013280].

## Author contributions

W.Q.: study design, data analysis and interpretation, manuscript writing, and editing; H.C.: study design, data analysis and interpretation, manuscript editing; A.B.D.: study design, data analysis and interpretation; S.L.: data analysis and interpretation; M.K.: data analysis and interpretation, manuscript editing; S.-I.L.: study design, data analysis and interpretation, manuscript editing, and supervision.

## Competing interests

The authors declare no competing interests.
