## [Peer Review File · Communications Medicine]

Reviewers' comments:

Reviewer #1 (Remarks to the Author):

I read this article with great pleasure and fluency. I appreciate the work done by the authors, and although we have conducted similar studies with XAI in our project, such a detailed presentation of the advantages of SHAP Tree explainer in observational studies will attract more people to use this tool for their research.

Some minor modifications are suggested.

1. Although further external validation is mentioned in the discussion section, I am curious why the results of the UK biobank data on the NHANES training model are not shown in it? I know that this result will receive a significant impact due to differences in data sources, but I think many readers will be interested in how large this impact really is? And are individuals able to use the published tools to assess the risk of death? If I get a risk value and the associated interpretation, how much can I trust the results? The interpretation may be correct for the group, but there may be potentially large risks involved for the individual.
2. In the discussion about reference ranges for laboratory tests, I think there is a risk in simply testing the appropriateness of reference ranges from the RRP. Is there a strong correlation between lab test values and age, and could these risks be due in large part to age? Would you be happy to show an interaction effect along with age?
3. When you use temporal validation, are the short term models in good agreement with the long term models? How do these differences arise and how are they explained? When clinical experts or individuals are faced with two black box models, how do they choose and how can they calibrate the different models for the current clinical setting?

some typo and editing issues need to be noted when revising.

Reviewer #2 (Remarks to the Author):

This is very well written paper. The suggested method is applied to a real data set and results are well explained and presented through graphs and figures. Some new insights into the associations between all-cause mortality and its predictors are presented.

I have only a few minor comments/concerns

This paper suggests an improvement on interpretable machine learning technique in predicting mortality. The method is applied to NHANES (1999-2014) data. The authors bring this fact (of improvement) only at the last sentence of the paper. Should not it also be a motivating factor?

At the beginning (e.g abstract and introduction), it may give the impression that the technique was developed as an improvement upon long existing linear models. This theme repeats but scantily throughout the paper .

Is all-cause mortality the central issue in epidemiology (the entire field) or it is an important issue?

Traditional regression fails utterly in case of complex non-linear situation. But it can be made to

work in several nonlinear or difficult cases (no match with MI though) using transformations, piecewise regression etc. (This is rightfully mentioned in Page 7, last paragraph; and it is appropriate to bring it there).

Anyway, the basic point is machine learning performs better, but may not have an obvious explanation, so—explainable methods, and here is an improvement on such methods. or something like that.....?

Figure 6- The text right below the graphs says p value < 0.0001 , while the value given with graph A is 0.0004 (perhaps a typo?). Does the exact test come out of 10x10 Table? Is it not too sparse? Is there a way to measure degree of association than only a p -value?

In the old regression model settings, once the coefficients are estimated then risk score or probability can be obtained for an individual patient by inputting his/her feature values. Can a physician get a similar value for his patients with this method and with the same easiness?

Reviewer #3 (Remarks to the Author):

In this study, the authors derive an explainable artificial intelligence system to predict mortality, using the NHANES dataset with nearly 50,000 individuals. They find that the IMPACT system achieved higher accuracy than linear models and neural networks, and that the explainable AI system discovered risk factors that were not included in traditional models. They further externally validate certain feature importance using the UK biobank, and develop an interactive website to prompt new research. The strengths of this paper are its novelty and contribution to the field of explainable AI. The external and temporal validation strategies are also strengths.

I have several suggestions that I believe would improve the explanation and description of this work:

- 1) In the abstract, please remove the phrases “accordingly adjust their lifestyle” and “help doctors give personalized treatment”. While the authors propose explainable AI, they do not propose identifying causality and developing actionable solutions. For example, it is likely that arm circumference (for example) is a surrogate for another more impactful risk factor.
- 2) Table 1 needs more explanation. How and in what populations are the scores for IMPACT presented? Is the same IMPACT model (with the same features) compared against the reference scores (Intermountain, Gagne, etc.)? Is it IMPACT tested on different datasets or time intervals?
- 3) More description into the NHANES dataset needs to be provided. In particular, two areas of concern arise from Figure 2: First, there is a heavy skew of samples towards the age 20 period. Why is this? How is sampling for NHANES performed? This raises the possibility of class imbalance in the dataset, which has implications into the performance metrics used (e.g. using another metric than AUROC for one-year mortality, such as AUPRC) and for the generalizability of the model.
- 4) Second, why are the number of samples for the mortality prediction so much higher than the number of samples within different age groups? Are the authors using the same patients at multiple time points in their mortality prediction? If this is the case, the authors should better describe how they account for repeated predictions, and whether IMPACT uses any methods to address this.
- 5) I would not characterize the UK Biobank as an external validation dataset. The features are completely different being used in the model. I struggle with whether the true purpose of the UK Biobank is to test the IMPACT approach in a novel dataset, or to externally validate the individual

features identified using the NHANES data.

6) In the last paragraph of the methods, the methodologic innovation needs to be better described. GBMs and Shapley plots have been extensively used in prior machine learning applications. Is the value of TreeExplainer the visualization interface? Is there another metric of explainability that is generated?

7) One worry I have is the individuality of predictions. For most individuals, age will be an extremely important predictor. Will the fact that age is a dominant predictor exclude the ability of Treeexplainer to identify other predictors that, while comparatively less important than age, still have considerable variability and importance between individual patients? Can the authors provide some metric of the variability of individual predictions and explainable features between patients?

8) I have concerns about the conclusions about laboratory reference intervals. It is still very unclear to me how IMPACT would be used to define alternative thresholds for specific patients. Can the authors provide an analysis that suggests what optimal reference ranges are?

9) Table 2 needs more explanation. Is the RRP the relative risk of mortality for “normal” ranges compared to abnormal ranges? For example, is what this is saying that the relative risk of one-year mortality for patients with a “normal” serum albumin 28.5% that of a patient with an abnormal albumin? How does one interpret negative values here? I also have some concern with the 100% values for ALT, which are suggestive of some issue with data missingness.

10) For section 3.3, I would be extremely cautious about describing the relative importance of different features over different time intervals until some of the concerns around sample construction (particularly the high representation of younger individuals) are addressed. Albumin, potassium, blood pressure, etc. mean something very different for a younger rather than an older individual.

11) While I agree with temporal validation, can the authors confirm that there was no age bias in more recent datasets? E.g. in longitudinal cohort studies, healthier individuals or younger individuals may live longer, and thus the distribution in a more recent test set is not representative of the general population. Is that potentially the case here?

MINOR

12) In the introduction, the “notable applications of AI in healthcare” are described after the sentence around supervised learning problems. Are all the examples supervised machine learning techniques? Are any of these (e.g. image-based detection) more representative of unsupervised learning?

Reviewer Comments and Revisions

We are especially grateful for the reviewers' careful critiquing of our research and their many suggestions for improvement. In response to their comments, we made significant changes that we believe address reviewers' concerns and considerably refine the document.

The following summarizes our changes. Reviewer 1's main concerns are addressed in (1), (2) and (3); Reviewer 2's in (3); and Reviewer 3's in (1), (2), (3) and (4).

- 1. Description of the temporal validation experiment design.** To test the generalizability of the IMPACT mortality risk scores, we perform "temporal" validation, i.e., testing the IMPACT mortality prediction models on hold-out data collected in recent cycles. As noted in item 4 below, different two-year cycles include completely different sets of individuals, with no overlap. Furthermore, to have similar base rates and age distributions in the training/testing and temporal validation sets, we use the samples from different collection cycles as the temporal validation set for different follow-up times. We added details about the temporal validation experiment design in *Supplementary Methods Section 6.4, "Comparing the predictive power of popular mortality risk scores and biological ages with IMPACT"*, and *Supplementary Figure 6*.
- 2. Explanation of the relative risk percentage for the reference intervals.** In the original manuscript, we use the relative risk percentage based on partial dependence plots (PDPs) to identify sub-optimal reference intervals for the laboratory features. PDPs show the marginal effect a set of features has on the predicted outcome of an ML model. For our models, the PDPs show the change in mortality risk for all values of a laboratory feature. We define the *relative mortality risk* as the average value of the model's predicted probability when we fix a specific feature to a given value divided by the average value of the model's predicted probability. Then, the *relative risk percentage* measures the relative risk of the feature values within the reference interval compared to the relative risk of all values. We note that we are not using the relative risk percentage to propose new reference intervals; instead, we are using it to assess whether existing reference intervals should be re-evaluated. We include details about the partial dependence plot and relative risk percentage in *Supplementary Methods Section 3.3, "Partial dependence plots and additional perspective to reference interval"*.
- 3. Additional external validation of the NHANES mortality prediction model using the UK Biobank dataset.** To validate the generalizability of the NHANES mortality prediction model and the explanations, we train a new NHANES 5-year mortality prediction model on the 51 overlapping features between NHANES and UK Biobank. We then validate the model's predictive performance, feature importance, and the relationship between the features and 5-year mortality *using the UK Biobank dataset*. The results are consistent with previous findings from the NHANES model. We also add the Spearman's correlation coefficient to test the consistency of the feature importance ranking. The new external validation results are incorporated in a new *Supplementary Appendix Section 1, "External validation of the NHANES mortality prediction model on the UK Biobank (UKB) dataset,"* and *Supplementary Figures 1 and 2*.
- 4. A clearer description of the NHANES dataset.** Initially, we would like to clarify that the NHANES 1999-2014 dataset we incorporated in our paper is *not* a longitudinal cohort study. Different samples were collected in each two-year cycle from different individuals. In terms of sample design, NHANES

oversampled younger and older subjects. Further, individuals of 80+ years were topcoded at 80 years of age. This accounts for the heavy skew of younger and older samples. We clarify our description of the NHANES data in *Supplementary Methods Section 1 “Data collection and processing”*.

We reply to their specific comments point-by-point below. The reviewer’s comments are shown in blue italics, with our replies in black. We include quotes from the revised manuscript in gray boxes and denote revised text in boldface type.

Referee expertise:

Referee #1: biomedical informatics, clinical big-data analysis, machine learning, heart disease

Referee #2: biostatistics, risk factors, epidemiology

Referee #3: clinician, machine learning, predictive analytics (including mortality)

Reviewers' comments:

Reviewer #1 (Remarks to the Author):

I read this article with great pleasure and fluency. I appreciate the work done by the authors, and although we have conducted similar studies with XAI in our project, such a detailed presentation of the advantages of SHAP Tree explainer in observational studies will attract more people to use this tool for their research.

Some minor modifications are suggested.

We greatly appreciate this positive feedback on our analyses. The reviewer’s suggestions are addressed below.

1. Although further external validation is mentioned in the discussion section, I am curious why the results of the UK biobank data on the NHANES training model are not shown in it? I know that this result will receive a significant impact due to differences in data sources, but I think many readers will be interested in how large this impact really is? And are individuals able to use the published tools to assess the risk of death? If I get a risk value and the associated interpretation, how much can I trust the results? The interpretation may be correct for the group, but there may be potentially large risks involved for the individual.

We thank the reviewer for raising excellent points. We add the results of the NHANES 5-year mortality prediction model tested on the UK biobank (UKB) data to a new Supplementary Appendix Section 1, “External validation of the NHANES mortality prediction model on the UK Biobank (UKB) dataset”. The NHANES and UKB datasets have different sets of features: there are 51 overlapping features between

them. As such, we are unable to directly test the NHANES mortality prediction model trained with 151 features on UKB data.

Instead, we train a *new* tree-based 5-year mortality prediction model on the NHANES dataset using the 51 overlapping features between NHANES and UKB. As shown in Supplementary Figure 2B, the classification accuracy on the UKB test set of the model trained on NHANES samples (AUROC=0.7780) and UKB samples (AUROC=0.7974) are close, which highlights the generalizability of the NHANES model. Supplementary Figure 1A shows the feature importances of the 51 features of the NHANES (51 features) and UKB models. *The SHAP values of both models are calculated using the same UKB samples as explicands (i.e., samples being explained) and baselines (i.e., background samples).* We observe that the top 20 most important features are largely consistent, where 14 features are the same for both models. The p-value of the Fisher's exact test ($p=0.0004$) shows that the overlap between the top 20 most important features of the NHANES and the UKB model is significant. The Spearman's correlation coefficient of the NHANES and UKB model's feature importance is 0.6969 ($p\text{-Value} < 0.0001$). Supplementary Figures 1B-G show the important results of the NHANES (51 features) model explained by the UKB samples: the SHAP main effect of red cell distribution width, serum albumin and serum uric acid, and the relative 5-year mortality risk of gamma glutamyl transferase, lymphocyte percent and serum albumin. The trends shown in these figures are consistent with previous findings from both the NHANES (151 features) and UKB (51 features) models.

We incorporate the new external validation results in *Results Section 3.2, "Discoveries from 5-year mortality prediction," lines 243-247 and lines 270-274; a new Supplementary Appendix Section 1, "External validation of the NHANES mortality prediction model on the UK Biobank (UKB) dataset," lines 2-19; and Supplementary Figures 1 and 2.*

Our external validation includes two aspects. The first aims to validate the entire IMPACT framework using a novel dataset by checking whether the explanations from a model trained on the NHANES dataset can also be found in a model trained on the UKB dataset. The second aims to test the generalizability of the mortality prediction model trained on the NHANES dataset.

The second one -- evaluating the generalizability of the mortality prediction model trained based on NHANES -- addresses this reviewer's comment.

Furthermore, we would like to validate whether the performance and explanations of the NHANES prediction model generalize to an unseen population (UKB). Training details and results are described in Supplementary Appendix 1 Section 1. Our external validation results show that the NHANES mortality prediction model generalizes well to the UKB dataset in terms of both mortality prediction performance and key relationships between features and mortality.

Supplementary Appendix 1 Section 1: External validation of the NHANES mortality prediction model on the UK Biobank (UKB) dataset

We aim to validate whether the performance and explanations of the NHANES mortality prediction model generalize to an unseen population (UKB). To do so, we train a *new* tree-based 5-year mortality prediction model on the NHANES dataset using the 51 overlapping features between NHANES and UKB. As shown in Supplementary Figure 2H, the classification accuracy on the UKB test set of the model trained on NHANES samples (AUROC = 0.7780) and UKB samples (AUROC = 0.7974) are close, which shows the generalizability of the NHANES model. Supplementary Figure 1A shows the feature importances of the 51 features of the NHANES (51 features) and UKB models. *The SHAP values of both models are calculated using the same UKB samples.* We observe that the top 20 most important features are largely consistent, with 14 features the same for both models. The p-value of the Fisher's exact test (p-value = 0.0004) shows that the overlap between the top 20 most important features of both models is significant. The Spearman's correlation coefficient of both models' feature importance is 0.6969 (p-value < 0.0001). Supplementary Figures 1B-G show noteworthy results of the NHANES (51 features) model explained by UKB samples: the SHAP main effect of red cell distribution width, serum albumin and serum uric acid, and the relative 5-year mortality risk of gamma glutamyl transferase, lymphocyte percent and serum albumin. The trends shown in these figures are consistent with previous findings from both the NHANES (151 features) and UKB (51 features) models. Additional validation results on the UKB dataset are presented in Supplementary Figure 2.

Supplementary Figure 1: External validation of the NHANES mortality prediction model on the UKB dataset. (A) Feature importance ranking of models trained on the NHANES (51 features) dataset and the UKB (51 features) dataset. The SHAP values are calculated using UKB samples. For each model, the figure shows the 20 most important features of prediction (ordered by importance). The purple line indicates that the feature is in the top 20 features of two models. Blue and red lines indicate the feature is in the top 20 features of one model but not in the top 20 features of the other. The p-value of the Fisher's exact test examines the overlap between the top 20 most important overlapping features in NHANES and UKB models (the contingency table in Supplementary Figure 2G). The Spearman's correlation coefficient is calculated using the feature importance of the overlapping features in NHANES and UKB. (*) represents a p-value < 0.001. (B)-(D) The main effect of red cell distribution width, urine albumin and serum uric acid on 5-year mortality of the model trained on the NHANES (51 features) dataset and explained using UKB samples. (E)-(G) The relative 5-year mortality risk of gamma glutamyl transferase, lymphocyte percent and serum albumin of the model trained on the NHANES (51 features) dataset and explained using UKB samples.**

Supplementary Figure 2: External validation of the NHANES mortality prediction model on the UKB dataset. (A) SHAP summary plot for the 5-year mortality prediction model trained on NHANES (51 features) dataset and explained using UKB samples. (B) The predictive performance of the models trained on the NHANES (51 features) and UKB (51 features) datasets. The AUROCs are calculated on the testing set by bootstrapping 1,000 times. (C)-(D) The main effect of serum albumin and platelet count on 5-year mortality of the model trained on the NHANES (51 features)

dataset and explained using UKB samples. (E)-(F) The relative 5-year mortality risk of alanine aminotransferase ALT on male and female samples of the model trained on the NHANES (51 features) dataset and explained using UKB samples. (G) The contingency table of the Fisher's exact test that evaluates the significance of the overlap between the top 20 most important overlapping features in the model trained on the NHANES (51 features) dataset and the model trained on the UKB (51 features) dataset. Both models are explained using UKB samples.

2. In the discussion about reference ranges for laboratory tests, I think there is a risk in simply testing the appropriateness of reference ranges from the RRP. Is there a strong correlation between lab test values and age, and could these risks be due in large part to age? Would you be happy to show an interaction effect along with age?

We thank the reviewer for the excellent question! Based on the question, we find that the description of the RRP (relative risk percentage) in the manuscript may not be sufficient. To clarify, the RRP is derived from *partial dependence plots*. The partial dependence function is calculated using the samples of all ages. So, the RRP is relatively robust to age. We add a rigorous definition of RRP to our revised manuscript, as follows.

We use partial dependence plots to show the change in mortality risk for all values of a laboratory feature. These plots show the marginal effect that a set of features has on the prediction of an ML model. The partial function f_S is estimated by:

$$f_S(x_S) = \frac{1}{n} \sum_{i=1}^n f(x_S, x_C^{(i)}).$$

In this formula, f is an ML model, and the S are features for which the partial dependence function should be plotted. In our study, S is the laboratory feature of interest and x_S is the given value of the feature. $x_C^{(i)}$ are actual feature values for the features of no interest in the test set, and n is the number of instances in the test set. The partial function tells us the average marginal effect on the prediction for given value(s) of features S . We extend the partial function to the relative mortality risk RR_S :

$$RR_S(x_S) = f_S(x_S) / \left(\frac{1}{n} \sum_{i=1}^n f(x^{(i)}) \right).$$

In other words, the relative mortality risk is defined as the average value of the model-predicted probability when we fix a specific feature to a given value divided by the average value of the model predicted probability. We further define the relative risk percentage (RRP) as follows:

$$RRP_S = \frac{\max(RR_S(x_S), x_S \text{ in RI}) - 1}{\max(RR_S(x_S)) - 1},$$

where RI stands for reference interval. We note that “age” is included in the $x_C^{(i)}$. The partial dependence plot of the laboratory feature is calculated using the samples of all ages. Thus, it is relatively robust to age.

Although PDPs are relatively robust to the effect of age because they average over many ages for a given feature, to explicitly show interaction effects as the reviewer mentioned, we visualize SHAP dependence plots of the laboratory features while coloring points by age, as follows:

Rebuttal Figure 1: (A)-(J): The SHAP dependence plots and the interactions of the laboratory features with age.

In Rebuttal Figure 1, we also show the Pearson correlation coefficients of the laboratory features and age to identify features for which a relationship with age exists. Then, to test if the SHAP values are due in large part to age, we treat the laboratory feature as the mediator of age and the SHAP value and perform the ‘Sobel test’ on the features significantly correlated with age. The Sobel test examines the significance of a mediation effect. The p-values are shown in Rebuttal Figure 1. We observe that the laboratory features all have significant effects on their respective SHAP values even given the effect of age. This confirms that the mortality risks (SHAP values) of these laboratory features are not due in large part to age.

We incorporated the above description of RRP in *Supplementary Methods Section 3.3, lines 96-110*:

We use *partial dependence plots* to show the change in mortality risk for all values of a laboratory feature. Partial dependence plots show the marginal effect a set of features has on the prediction of an ML model. The partial function f_S is estimated by:

$$f_S(x_S) = \frac{1}{n} \sum_{i=1}^n f(x_S, x_C^{(i)}).$$

In this formula, f is an ML model, and the S are features for which the partial dependence function should be plotted. In our study, S is the laboratory feature of interest and x_S is the given value of the feature. $x_C^{(i)}$ is actual feature values for the features of no interest in the test set, and n is the number of instances in the test set. The partial function tells us the average marginal effect on the prediction for given value(s) of features S . We extend the partial function to the relative mortality risk RR_S :

$$RR_S(x_S) = f_S(x_S) / \left(\frac{1}{n} \sum_{i=1}^n f(x^{(i)}) \right).$$

In other words, the relative mortality risk is defined as the average value of the model's predicted probability when we fix a specific feature to a given value divided by the average value of the model's predicted probability. We further define the relative risk percentage (RRP) as follows:

$$RRP_S = \frac{\max(RR_S(x_S), x_S \text{ in RI}) - 1}{\max(RR_S(x_S)) - 1},$$

where RI stands for reference interval.

3. When you use temporal validation, are the short term models in good agreement with the long term models? How do these differences arise and how are they explained? When clinical experts or individuals are faced with two black box models, how do they choose and how can they calibrate the different models for the current clinical setting?

Thank you for raising this important question. Perhaps our use of the term “temporal validation” in our original manuscript was insufficiently clear. The “temporal validation” and the “comparison of the short-term model and the long-term model” refer to different experiments and have different purposes. Here, we first clarify the difference between the temporal validation and the models trained with different follow-up times.

In this study, “temporal validation” refers to demonstrating that each model works similarly on patient data collected during different time periods used as a hold-out dataset. To accomplish this, we hold out some data collected in the recent cycles to evaluate the generalizability of the mortality scores for samples collected at different times (Table 1, Figure 9B). Thus, “temporal validation” does not compare across shorter-term and longer-term models, but within each model.

Specifically, for 1-year mortality prediction, we use samples collected in 1999-2012 as the training/testing set and samples collected in 2013-2014 as the temporal validation set. For 5-year mortality prediction, we

use samples collected in 1999-2008 as the training/testing set and those collected in 2009-2014 as the temporal validation set. For 10-year mortality prediction, we use samples collected in 1999-2000 as the training/testing set and those collected in 2001-2014 as the temporal validation set. Samples that are not included in the temporal validation set are randomly split into 80% for training and 20% for testing. Hence, samples in the temporal validation set are not used for model training. They are used only to evaluate the models' performance.

We agree with the reviewer that understanding performance across models is important for future clinical utility. To compare the short- and long-term models, we compare the feature importance of 1-year, 3-year, 5-year, and 10-year mortality prediction models. As discussed in Results Section 3.3, Figure 7A shows the top 20 most important features and relative importance of input features in IMPACT's 1-year, 3-year, 5-year, and 10-year mortality prediction models. Feature importance rankings change significantly across these four models. Some features are important for all four (e.g., age, RDW, and urine albumin level). Some become more important over time (e.g., platelet count, with an importance ranking of 75 for the 1-year model and 12 for the 10-year model). Some become less important over time (e.g., serum potassium, with an importance ranking of 17 for the 1-year model and 42 for the 10-year model). Furthermore, the relationship between each feature and mortality may change for models that use different follow-up times. For instance, Figures 7B-C show the SHAP values for serum potassium in IMPACT's 1-year and 5-year mortality prediction models. For the 1-year model, hyperkalemia (high potassium) has a higher mortality risk than hypokalemia (low potassium). For the 5-year model, hypokalemia has the same or higher mortality risk than hyperkalemia.

These results highlight the differences between short and longer term models. These differences indicate that the contributions of the risk predictors to shorter and longer term mortality prediction differs. They further suggest that the influences of risk predictors on shorter and longer term mortality differ.

Future application of these models for clinical practice is an important consideration. Perhaps those important features shared across models can serve as a general baseline with relevance to most patients, while those that are differentially important in the different models can be applied based on specific needs and individual situation. Intuitively, it would seem that patients with predictors suggesting a high 1-year mortality risk would want to treat those predictors more acutely and aggressively than patients with a high 5- or 10-year mortality risk. However, since we are not clinicians, we have intentionally not attempted to recommend clinical use of these tools in the manuscript.

We clarify the temporal validation in *Supplementary Methods Section 6.4, "Comparing the predictive power of popular mortality risk scores and biological ages with IMPACT," lines 248-260*:

We perform temporal validation to show the generalizability of the IMPACT-20 risk scores on data collected at different time periods. To have similar base rates and age distributions in the test set and temporal validation set, we use the samples from different collection cycles as the temporal validation set for different follow-up times. For 1-year mortality prediction, we use the samples collected in 1999-2012 as the training/testing set and the samples collected in 2013-2014 as the temporal validation set. For 5-year mortality prediction, we use the samples collected in 1999-2008

as the training/testing set and those collected in 2009-2014 as the temporal validation set. For 10-year mortality prediction, we use the samples collected in 1999-2000 as the training/testing set and those collected in 2001-2014 as the temporal validation set. With respect to the 5-year mortality risk scores, samples that are not included in the temporal validation set are randomly split into 80% for training and 20% for testing. The sample size, number of deceased samples, and histogram of age in the training set, with the testing and temporal validation sets, are shown in Supplementary Figure 6.

some typo and editing issues need to be noted when revising.

Thank you. We reviewed the manuscript to correct typos and resolve editing issues.

Reviewer #2 (Remarks to the Author):

This is very well written paper. The suggested method is applied to a real data set and results are well explained and presented through graphs and figures. Some new insights into the associations between all-cause mortality and its predictors are presented.

I have only a few minor comments/concerns

We appreciate this positive feedback on our analysis. We address the reviewer's comments and concerns below.

This paper suggests an improvement on interpretable machine learning technique in predicting mortality. The method is applied to NHANES (1999-2014) data. The authors bring this fact (of improvement) only at the last sentence of the paper. Should not it also be a motivating factor?

We thank the reviewer for raising an excellent point. We highlight this motivating factor in the original abstract by saying that "IMPACT's unique strength is the *explainable* prediction, which provides insights into the complex, non-linear relationships between mortality and individual's features while maintaining high model accuracy and the expressive power to capture complex relationships." To further highlight this motivation, we added a sentence in the *Introduction*, lines 69-71:

In this paper, we present the IMPACT (Interpretable Machine learning Prediction of All-Cause mortality) framework (Figure 1), which improves the interpretability of complex machine learning models for mortality prediction.

At the beginning (e.g abstract and introduction), it may give the impression that the technique was developed as an improvement upon long existing linear models. This theme repeats but scantily throughout the paper .

Is all-cause mortality the central issue in epidemiology (the entire field) or it is an important issue?

We appreciate your raising this point. In response, we revised the statement in the *Introduction*, lines 50-51:

Identification of risk factors and prediction of all-cause mortality have long been important issues in epidemiology.

Traditional regression fails utterly in case of complex non-linear situation. But it can be made to work in several nonlinear or difficult cases (no match with MI though) using transformations, piecewise regression etc. (This is rightfully mentioned in Page 7, last paragraph; and it is appropriate to bring it there).

Anyway, the basic point is machine learning performs better, but may not have an obvious explanation, so—explainable methods, and here is an improvement on such methods. or something like that.....?

This is a great suggestion. In the original Introduction, we mentioned the advantages of the complex ML models and the importance of explanation by writing that: “The field of artificial intelligence (AI) has seen significant advances in supervised learning problems, which involve predicting an outcome variable (e.g., all-cause mortality) based on a set of features (e.g., individual-level characteristics). A major obstacle to the adoption of AI applications in healthcare is that many of them are considered “black box,” which refers to their lack of interpretability. The inability to understand why a model makes a prediction is especially harmful in healthcare applications, where the patterns a model discovers can be even more important than its predictive accuracy. This is especially true in epidemiology, which aims to identify important variables to guide public health policy or detect risk predictors that warrant further study. To address this need, we turn to a variety of techniques to help us better understand complex ML models from the emerging area of explainable AI (XAI).”

In response to the reviewer’s comment, we further highlight the motivation by adding two sentences in the *Introduction*, lines 69-71 and lines 90-91:

In this paper, we present the IMPACT (Interpretable Machine learning Prediction of All-Cause morTality) framework (Figure 1), which improves the interpretability of complex machine learning models for mortality prediction.

The IMPACT framework can also be applied to other health outcomes and diseases to improve the predictive accuracy and interpretability of complex ML models in epidemiological studies.

Figure 6- The text right below the graphs says p value < 0.0001 , while the value given with graph A is 0.0004 (perhaps a typo?). Does the exact test come out of 10×10 Table? Is it not too sparse? Is there a way to measure degree of association than only a p -value?

Thank you for this excellent question. We note that (***) represents a p -value < 0.001 , not 0.0001 as we mentioned in the caption of Figure 6. Here, the p -value is from the Fisher's exact test of the overlap between the top 20 most important overlapping features in the NHANES and UK Biobank models. The contingency table of the Fisher's exact test is as follows:

(F)

Top 20 feature set			
	NHANES	Not NHANES	Total
UK Biobank	14	6	20
Not UK Biobank	6	25	31
Total	20	31	51

Rebuttal Figure 2: The contingency table of the Fisher's exact test, which evaluates the significance of the overlap between the top 20 most important overlapping features in the NHANES and UK Biobank models.

The table is also shown in Supplementary Figure 9F. To supplement Fisher's exact test, we calculate Spearman's correlation of the feature importance of the overlapping features in NHANES and UKB. The Spearman's correlation coefficient is $r_s = 0.6654$, p -value < 0.0001 , which shows the significant positive correlation between the feature importance rankings of the overlapping features in NHANES and UKB.

We add the Spearman's correlation coefficient to *Results Section 3.2, "Discoveries from 5-year mortality prediction," lines 255-257* and to *Figure 6A*:

The Spearman's correlation coefficient of the NHANES and UKB model's feature importance is 0.6654 (p -value < 0.0001), showing the significant positive correlation between the ranking of the overlapping features in NHANES and UKB.

In the old regression model settings, once the coefficients are estimated then risk score or probability can be obtained for an individual patient by inputting his/her feature values. Can a physician get a similar value for his patients with this method and with the same easiness?

We thank the reviewer for the excellent question. The 5-year IMPACT mortality risk scores using the top 20 features are available on our interactive website, <https://suinleelab.github.io/IMPACT>. We also uploaded the trained models and feature orders to the Github repository (<https://github.com/suinleelab/IMPACT>). Researchers can easily load and evaluate the models on their own data using Python.

Reviewer #3 (Remarks to the Author):

In this study, the authors derive an explainable artificial intelligence system to predict mortality, using the NHANES dataset with nearly 50,000 individuals. They find that the IMPACT system achieved higher accuracy than linear models and neural networks, and that the explainable AI system discovered risk factors that were not included in traditional models. They further externally validate certain feature importance using the UK biobank, and develop an interactive website to prompt new research. The strengths of this paper are its novelty and contribution to the field of explainable AI. The external and temporal validation strategies are also strengths.

I have several suggestions that I believe would improve the explanation and description of this work:

We appreciate the reviewer's very thorough and helpful feedback on our analysis, which has given us the opportunity to improve our manuscript as noted below.

1) In the abstract, please remove the phrases "accordingly adjust their lifestyle" and "help doctors give personalized treatment". While the authors propose explainable AI, they do not propose identifying causality and developing actionable solutions. For example, it is likely that arm circumference (for example) is a surrogate for another more impactful risk factor.

The reviewer makes an excellent point! In light of it, we revised the sentence in the *Plain language summary*, lines 46-47:

The interpretable risk scores can help both individuals improve health awareness and gain insight into their health status and health professionals identify high-risk individuals.

2) Table 1 needs more explanation. How and in what populations are the scores for IMPACT presented? Is the same IMPACT model (with the same features) compared against the reference scores (Intermountain, Gagne, etc.)? Is it IMPACT tested on different datasets or time intervals?

We completely agree with the reviewer that we need to clarify the explanation in Table 1. Each row in Table 1 compares the AUROCs between an existing mortality score reported in its original paper and the IMPACT-20 model tested for the corresponding follow-up time and age ranges in the NHANES dataset. Here, IMPACT-20 means the IMPACT model when the top 20 features were used (listed in Supplementary Tables 2 and 3); as shown in Figures 9A-B, with 20 features, the IMPACT model achieves nearly maximal performance with all features.

Here is our reasoning for such a comparison. Since not all features used in the popular mortality risk scores and biological ages are included in the NHANES dataset (except for the Intermountain risk score; see below), it would not be fair to compare existing mortality scores and biological ages computed based on a partial set of features with the IMPACT model based on the NHANES dataset. Therefore, we chose

to show the AUROCs reported in the original papers. Since the AUROCs are *not* sensitive to the base rate, we assume that these scores would be consistent across different datasets if risk scores and biological ages generalize well. Then, we performed a direct comparison on the NHANES dataset for the Intermountain risk score because that dataset contains all features the Intermountain risk score uses.

As we mention in Supplementary Methods Section 6.3, “Intermountain mortality risk score,” all features used in the Intermountain risk scores are included in our NHANES dataset. Thus, in Results Section 3.5, “Highly accurate and efficient interpretable mortality risk scores,” we calculate the Intermountain mortality risk score using NHANES samples and compare it with the IMPACT mortality risk scores. The classification performance of our IMPACT mortality risk score using the same number of features (14) exceeds that for the Intermountain risk score. To remain consistent with other risk scores and biological ages, we do not incorporate the AUROC of the Intermountain risk score tested on our NHANES dataset in Table 1. The AUROC of the Intermountain risk score in Table 1 is the result from the original paper.

While we attempted to address the reviewer’s question, “Is IMPACT tested on different datasets or time intervals?”, we find that our description of the test and temporal validation sets may not have been sufficient.

We perform temporal validation to show the generalizability of the IMPACT-20 risk scores on held-out data; in this case, they are the data collected at different time periods in the NHANES study. To have similar age distributions in the test and temporal validation sets, we use the samples from different collection cycles as the temporal validation set for different follow-up times. For 1-year mortality prediction, we use the samples collected in 1999-2012 as the training/testing set and the samples collected in 2013-2014 as the temporal validation set. For 5-year mortality prediction, we use the samples collected in 1999-2008 as the training/testing set and the samples collected in 2009-2014 as the temporal validation set. For 10-year mortality prediction, we use the samples collected in 1999-2000 as the training/testing set and the samples collected in 2001-2014 as the temporal validation set. In the training/testing set, we use 80% samples for training and 20% samples for testing. The sample size, the number of deceased samples, and the histogram of age in the training set, with the testing set and the temporal validation set, are shown in Supplementary Figure 6.

We incorporated the preceding description into the *Introduction*, lines 81-83; the caption of *Table 1*, a new *Supplementary Methods Section 6.4* “*Comparing the predictive power of popular mortality risk scores and biological ages with IMPACT*,” lines 232-264 alongside *Supplementary Figure 6*, and *Supplementary Table 3*:

In Table 1, we compare the AUROCs between an existing mortality score or a biological age as reported in the original paper and the IMPACT-20 model tested for the corresponding follow-up time and age ranges in the NHANES dataset. We find that IMPACT risk scores (Supplementary Methods 6.4) have higher predictive power than popular mortality risk scores and biological ages.

	Task	Age	AUROC	AUROC of IMPACT-20	AUROC of IMPACT-20 (temporal validation)
Mortality risk scores					
Intermountain [17]	1-year mortality	18+	0.84	0.92	0.88
Gagne Index [10]	1-year mortality	65+	0.79	0.85	0.85
Intermountain [17]	5-year mortality	18+	0.87	0.89	0.88
Prognostic score [11]	5-year mortality	40-70	Male: 0.80	Male: 0.85	Male: 0.80
			Female: 0.79	Female: 0.83	Female: 0.80
Schonberg Index [46]	5-year mortality	65+	0.75	0.80	0.83
Biological ages					
Horvath DNAm Age [18, 23]	10-year mortality	21-84	0.56	0.90	0.89
Hannum DNAm Age [15, 23]	10-year mortality	21-84	0.57	0.90	0.89
DNAm PhenoAge [23]	10-year mortality	21-84	0.62	0.90	0.89
Phenotypic Age [23, 24]	10-year mortality	20-85	0.88	0.90	0.89

Table 1: Comparing the AUROCs between an existing mortality score or a biological age as reported in the original paper and the IMPACT-20 model tested for the corresponding follow-up time and age ranges in the NHANES dataset. The “AUROC” column shows the AUROCs reported in the original paper. The “AUROC of IMPACT-20” column shows the performance of IMPACT models trained with the selected top 20 features (Supplementary Tables 2 and 3). The “AUROC of IMPACT-20 (temporal validation)” column shows the performance of the IMPACT-20 models evaluated on the temporal validation set (Supplementary Methods 6.4).

[10] Joshua J Gagne et al. “A combined comorbidity score predicted mortality in elderly patients better than existing scores”. In: *Journal of clinical epidemiology* 64.7 (2011), pp. 749–759.

[11] Andrea Ganna and Erik Ingelsson. “5 year mortality predictors in 498 103 UK Biobank participants: A prospective population-based study”. In: *The Lancet* 386.9993 (2015), pp. 533–540. issn: 1474547X. doi:10.1016/S0140-6736(15)60175-1. Url: [http://dx.doi.org/10.1016/S0140-6736\(15\)60175-521-1](http://dx.doi.org/10.1016/S0140-6736(15)60175-521-1).

[15] Gregory Hannum et al. “Genome-wide methylation profiles reveal quantitative views of human aging rates”. In: *Molecular cell* 49.2 (2013), pp. 359–367.

[17] Benjamin D Horne et al. “Exceptional mortality prediction by risk scores from common laboratory tests”. In: *The American journal of medicine* 122.6 (2009), pp. 550–558.

[18] Steve Horvath. “DNA methylation age of human tissues and cell types”. In: *Genome biology* 14.10 (2013), pp. 1–20.

[23] Morgan E Levine et al. “An epigenetic biomarker of aging for lifespan and healthspan”. In: *Aging (Albany NY)* 10.4 (2018), p. 573.

[24] Zuyun Liu et al. “A new aging measure captures morbidity and mortality risk across diverse subpopulations from NHANES IV: a cohort study”. In: *PLoS medicine* 15.12 (2018), e1002718.

[46] Mara A Schonberg et al. “Index to predict 5-year mortality of community-dwelling adults aged 65 and older using data from the National Health Interview Survey”. In: *Journal of general internal medicine* 24.10 (2009), p. 1115.

Supplementary Methods 6.4: Comparing the predictive power of popular mortality risk scores and biological ages with IMPACT

Since not all features used in the popular mortality risk scores and biological ages are included in the NHANES dataset (except for Intermountain risk scores; see Supplementary Methods 6.3), it would not be fair to compare the existing mortality scores and biological ages computed based on

a partial set of features with the IMPACT model based on the NHANES dataset. Therefore, we chose to show the AUROCs reported in the original papers. As the AUROCs are *not* sensitive to the base rate, we assume that these scores would be consistent among different datasets if the risk scores and biological ages generalize well.

Table 1 compares the AUROCs between an existing mortality score or a biological age as reported in the original paper and the IMPACT-20 model tested for the corresponding follow-up time and age ranges in the NHANES dataset. Here, IMPACT-20 means the IMPACT model when the top 20 features were used; we chose 20 features because in Figures 9A-B, the IMPACT model with 20 features obtains an AUROC that is almost the same as the performance of the model using all features, and using fewer than 20 features leads to a dramatic decline in accuracy.

To get the top 20 most important features for 1-year and 10-year mortality predictions, we repeat the same mortality risk scores training and recursive feature elimination process for 1-year and 10-year predictions. We perform temporal validation to show the generalizability of the IMPACT-20 risk scores on data collected at different time periods. To have similar base rates and age distributions in the test set and temporal validation set, we use the samples from different collection cycles as the temporal validation set for different follow-up times. For 1-year mortality prediction, we use the samples collected in 1999-2012 as the training/testing set and the samples collected in 2013-2014 as the temporal validation set. For 5-year mortality prediction, we use the samples collected in 1999-2008 as the training/testing set and those collected in 2009-2014 as the temporal validation set. For 10-year mortality prediction, we use the samples collected in 1999-2000 as the training/testing set and those collected in 2001-2014 as the temporal validation set. With respect to the 5-year mortality risk scores, samples that are not included in the temporal validation set are randomly split into 80% for training and 20% for testing. The sample size, number of deceased samples, and histogram of age in the training set, with the testing and temporal validation sets, are shown in Supplementary Figure 6. In Table 1, the “AUROC” column shows the AUROCs reported in the original paper. The “AUROC of IMPACT-20” column shows the performance of 1-year, 5-year, and 10-year IMPACT models trained with the selected top 20 features (listed in Supplementary Tables 2 and 3). The IMPACT-20 models are trained on samples of all ages and evaluated on the samples within the same age range in the original paper. We bootstrap the test set and the temporal validation set for 1,000 times when measuring the AUROCs.

(A)

Follow-up time	Training set (80%) + Testing set (20%)				Temporal validation set			
	Collection cycles	Number of samples	Number of deaths	Base rate	Collection cycles	Number of samples	Number of deaths	Base rate
1-year	1999-2012	41,179	524	1.27%	2013-2014	6,082	53	0.81%
5-year	1999-2008	28,820	2,247	7.80%	2009-2014	7,034	827	11.76%
10-year	1999-2000	5,444	931	17.10%	2001-2014	16,542	4,364	26.38%

Supplementary Figure 6: (A) Population characteristics of the training/testing and temporal validation sets with different follow-up times. (B)-(G) Histograms of age in the training/testing set and temporal validation set with different follow-up times.

Importance Ranking	IMPACT-20 (1-year mortality prediction)	IMPACT-20 (10-year mortality prediction)
1	Age	Age
2	Albumin, serum (g/L)	Albumin, urine (ug/mL)
3	Albumin, urine (ug/mL)	Blood lead (umol/L)
4	Lymphocyte percent (%)	General health condition
5	Blood lead (umol/L)	Albumin, serum (g/L)
6	Education Level - Adults 20+	Arm Circumference (cm)
7	Red cell distribution width (%)	Red cell distribution width (%)
8	Cholesterol, serum (mmol/L)	Chloride, serum (mmol/L)
9	Blood mercury, total (ug/L)	Education Level - Adults 20+
10	General health condition	Blood cadmium (nmol/L)
11	Red blood cell count (million cells/uL)	Creatinine, serum (umol/L)
12	Basophils percent (%)	Received Hepatitis B 3 dose series
13	Require special healthcare equipment (0-No, 1-Yes)	Self-reported greatest weight (pounds)
14	Arm Circumference (cm)	Body Mass Index (kg/m**2)
15	Upper Arm Length (cm)	Systolic: Blood pres (2nd rdg) mm Hg
16	Blood cadmium (nmol/L)	Mean cell hemoglobin (pg)
17	Chloride, serum (mmol/L)	Gamma glutamyl transferase (U/L)
18	Avg # alcoholic drinks/day - past 12 mos	Potassium, serum (mmol/L)
19	Systolic: Blood pres (1st rdg) mm Hg	Blood mercury, total (ug/L)
20	Blood urea nitrogen (mmol/L)	How do you consider your weight?

Supplementary Table 3: Selected top 20 features of the 1-year and the 10-year mortality risk scores.

3) More description into the NHANES dataset needs to be provided. In particular, two areas of concern arise from Figure 2: First, there is a heavy skew of samples towards the age 20 period. Why is this? How is sampling for NHANES performed? This raises the possibility of class imbalance in the dataset, which has implications into the performance metrics used (e.g. using another metric than AUROC for one-year mortality, such as AUPRC) and for the generalizability of the model.

We appreciate the excellent question. First, we entered an incorrect histogram of age in the manuscript and apologize for the mistake. The following is the corrected histogram of age of the samples included in our study:

Rebuttal Figure 3: The histogram of age.

Based on the reviewer’s question about how NHANES performs sampling, we refer to the sample design from the “National Health and Nutrition Examination Survey: Plan and Operations, 1999–2010”¹: “*The National Health and Nutrition Examination Survey (NHANES) is a program of studies designed to assess the health and nutritional status of adults and children in the United States. In 1999, the survey became a continuous program that has a changing focus on a variety of health and nutrition measurements to meet emerging needs. Data was released to the public in 2-year cycles. The sample size in a 12-month period was approximately 5,000 individuals from 15 different county locations selected from a sampling frame that included all 50 states and the District of Columbia. **The design of the sample changed periodically. Oversampled subgroups for 1999–2006 included non-Hispanic black persons, Mexican-American persons, low-income white persons (beginning in 2000), adolescents aged 12–19, and persons aged 70 and over. Oversampled subgroups for 2007–2010 included all Hispanic persons, non-Hispanic black persons, low-income white persons, and persons aged 80 and over.***”

In our analysis (both for the original and revised manuscripts), we exclude participants under age 18 because they are not eligible for the public release of mortality data. In addition, in the raw data, individuals 85 and over are topcoded at 85 years of age in NHANES 1999-2006, and individuals 80 and over are topcoded at 80 years of age in NHANES 2007-2014. To maintain consistency, we topcode individuals 80 and over at 80 years of age and indicate in the manuscript’s age distribution plots that the last bin corresponds to 80 or older (80+). Therefore, the NHANES sample design and the data preprocessing (age labeling) heavily skew samples towards the age 20 and 80 periods, respectively.

To examine whether the large number of young samples influences the performance and explanations of the models, we subsample the young samples to produce a more “age balanced” dataset and replicate the key results. Specifically, we divide the samples into 20 bins based on age. Then, we subsample the samples in the first bin (ages 18-21.1) to the average number of samples in the other 18 bins except the oldest one, split the data into 80% for training and 20% for testing, and retrain the models. We also test

¹ https://www.cdc.gov/nchs/data/series/sr_01/sr01_056.pdf

the original models' performance on this age-balanced data to show the generalizability of the original models. The new histograms of age and models' performance are:

(E)

Follow-up time	Original model test on original data		Age balance model test on age balance data		Original model test on age balance data	
	Base rate	AUROC	Base rate	AUROC	Base rate	AUROC
1-year	0.0122	0.9219	0.0130	0.9152	0.0130	0.9175
3-year	0.0458	0.9032	0.0487	0.9014	0.0487	0.9016
5-year	0.0857	0.8941	0.0913	0.8938	0.0913	0.8929
10-year	0.2408	0.9195	0.2594	0.9155	0.2594	0.9155

Rebuttal Figure 4: (A)-(D) Histogram of age after the sub-sampling of young samples. (E) The AUROCs and base rates of the original models and the age balance models.

The following Rebuttal Figure 5 compares feature importance of the original 5-year mortality prediction model and the “age balance” 5-year mortality prediction model and shows some key results of the latter:

Rebuttal Figure 5: (A) Relative importance of input features in the original 5-year mortality prediction model and the ‘age balance’ 5-year mortality prediction model. (B)-(E) The key SHAP plots of the age balance 5-year mortality prediction model. (F)-(K) The partial dependence plots of the age balance 5-year mortality prediction model.

From the figure, we observe that the AUROCs, feature importance, and key results are nearly identical to those of the original models. A large number of young samples do not have a strong influence on the performance of the explanations of the models.

We thank the reviewer for suggesting using AUPRC in addition to AUROC. We add the AUPRCs to the manuscript (Supplementary Figure 7). We find that tree-based models outperform both linear models and neural networks in seven of the tasks we consider, which is consistent with findings from the AUROC results in Figure 3A.

We incorporated the description of the sample design to *Supplementary Methods Section 1, “Data collection and processing,”* lines 9-13, and we add the AUPRC to a new *Supplementary Figure 7*:

The National Health and Nutrition Examination Survey (NHANES) from the National Center for Health Statistics (NCHS) conducts interviews and physical examinations to assess the health and nutrition data for all ages in the United States. The interviews include demographic, socioeconomic, dietary, and health-related questions. The examinations include medical, dental, physiological measurements, and laboratory tests administered by highly trained medical personnel. Since 1999, data were collected and released at 2-year intervals. Each year NHANES examines a nationally representative sample of roughly 5,000 individuals across the United States. **The design of the sample changed periodically. Oversampled subgroups for 1999–2006 included non-Hispanic black persons, Mexican-American persons, low-income white persons (beginning in 2000), adolescents aged 12–19, and persons aged 70 and over. Oversampled subgroups for 2007–2010 included all Hispanic persons, non-Hispanic black persons, low-income white persons, and persons aged 80 and over.** In this study, we include NHANES data sampled between 1999 and 2014. All-cause mortality is ascertained by a linked NHANES mortality file that provides follow-up mortality data from the date of survey participation through December 31, 2015. **We exclude participants under age 18 because they are not eligible for public release mortality data.**

Improving prediction power

AUPRC on different mortality prediction tasks

	Logistic Regression	Gradient Boosted Trees	Neural Network
1-year mortality	0.1405 — ***	0.2307 — ***	0.1016
3-year mortality	0.3787 — ***	0.4397 — ***	0.3507
5-year mortality	0.5131 — ***	0.5464 — ***	0.4838
10-year mortality	0.7980 — ***	0.8212 — ***	0.7066
Age < 40	0.0441	0.1047 — *	0.0423
40 ≤ Age < 65	0.3436	0.3823 — ***	0.2931
65 ≤ Age < 80	0.5263 — **	0.5790 — ***	0.4717
Age ≥ 80	0.7447	0.7071	0.6766

Supplementary Figure 7: The area under the precision-recall curve (AUPRC) of gradient boosted tree models outperforms both linear models and neural networks for seven of our prediction models. (*) represents a p-value < 0.001, (**) represents a p-value < 0.01, and (*) represents a p-value < 0.05. P-values are computed using bootstrap resampling over the tested time points while measuring the difference in area between the curves.**

4) Second, why are the number of samples for the mortality prediction so much higher than the number of samples within different age groups? Are the authors using the same patients at multiple time points in their mortality prediction? If this is the case, the authors should better describe how they account for repeated predictions, and whether IMPACT uses any methods to address this.

We appreciate the reviewer’s highlighting this point of confusion. NHANES 1999-2014 is *not* a longitudinal cohort study. In the NHANES 1999-2014, different samples were collected in each two-year cycle from different individuals. Therefore, different cycles include completely different sets of individuals, with no overlap.

As the following figure shows, when we predict mortality within different age groups, we fix the follow-up time to 5-years and divide all samples for 5-year mortality prediction into four sets based on age: age<40, 40<=age<65, 65<=age<80 and age>=80.

Rebuttal Figure 6: The number of samples in different age groups.

In other words, the numbers of samples within each age group add up to the number of samples for the 5-year mortality prediction. For the mortality prediction with different follow-up times, we use samples of all ages. Therefore, it is natural that the number of samples for the mortality prediction exceeds the number of samples within different age groups, and we are not using the same patients at multiple time points.

We clarify this point in *Methods Section 2.1, “Data cohorts,” lines 100-102*:

For mortality prediction with different follow-up times, we use samples of all ages. For different age groups, we fix the follow-up time to predict 5-year mortality and divide all samples for 5-year mortality prediction into four sets based on age.

5) I would not characterize the UK Biobank as an external validation dataset. The features are completely different being used in the model. I struggle with whether the true purpose of the UK Biobank is to test the IMPACT approach in a novel dataset, or to externally validate the individual features identified using the NHANES data.

We thank the reviewer for raising an excellent point. First, we clarify that the features used in the NHANES and UK Biobank (UKB) models are not completely different. The UKB model uses 51 overlapping features between the NHANES and UKB datasets. And, as shown in Figure 6, most of the important features that lead to new interesting findings (e.g., presented in Figures 3, 4 and 5) are included in these 51 features. Furthermore, with respect to characterizing UKB as an external validation dataset, we add the results of externally evaluating the NHANES 5-year mortality prediction model on UKB data.

To clarify, we have two types of external validation. The first aims to validate the entire IMPACT framework -- starting from training the prediction model to measuring prediction accuracy and generating explanations -- in a novel dataset by checking whether the explanations from a model trained on the NHANES dataset can also be found in a model trained on UKB. For validation, we train two separate models on NHANES (151 features) and UKB (51 features). The results are included in our initial submission (Figure 6).

The second type of validation aims to test the generalizability of the mortality prediction model *trained on the NHANES dataset*. We would like to validate whether the performance and explanations of the NHANES mortality prediction model generalize to an unseen population (UKB). We add the new external validation results to a new Supplementary Appendix 1 Section 1, “External validation of the NHANES mortality prediction model on the UK Biobank (UKB) dataset.” Specifically, we train a *new* tree-based 5-year mortality prediction model on NHANES using the 51 overlapping features between NHANES and UKB. As shown in Supplementary Figure 2B, the classification accuracy on the UKB test set of the model trained on NHANES samples (AUROC=0.7780) and UKB samples (AUROC=0.7974) are close, which shows the generalizability of the NHANES model. Supplementary Figure 1A shows the feature importances of the 51 features of NHANES (51 features) and UKB. *The SHAP values of both models are calculated using the same UKB samples.* We observe that the top 20 most important features are largely consistent, where 14 features are the same for both models. The p-value of the Fisher’s exact test ($p=0.0004$) shows that the overlap between the top 20 most important features of NHANES and UKB is significant. The Spearman’s correlation coefficient of the NHANES and UKB model’s feature importance is 0.6969 ($p\text{-Value} < 0.0001$). Supplementary Figures 1B-G show the important results of the NHANES (51 features) model explained by the UKB samples: the SHAP main effect of red cell distribution width, serum albumin and serum uric acid, and the relative 5-year mortality of gamma glutamyl transferase, lymphocyte percent and serum albumin. The trends shown in these figures are consistent with the previous findings from both NHANES (151 features) and UKB (51 features).

We incorporate the new external validation results to *Results Section 3.2, “Discoveries from 5-year mortality prediction,” lines 243-247 and lines 270-274; a new Supplementary Appendix 1 Section 1, “External validation of the NHANES mortality prediction model on UK Biobank (UKB) dataset,” lines 2-19, and Supplementary Figures 1 and 2.*

Our external validation includes two aspects. The first aims to validate the entire IMPACT framework using a novel dataset by checking whether the explanations from a model trained on the NHANES dataset can also be found in a model trained on the UKB dataset. The second aims to test the generalizability of the mortality prediction model trained on the NHANES dataset.

Furthermore, we would like to validate whether the performance and explanations of the NHANES prediction model generalize to an unseen population (UKB). Training details and results are described in Supplementary Appendix 1 Section 1. Our external validation results show that the NHANES mortality prediction model generalizes well to the UKB dataset in terms of both mortality prediction performance and key relationships between features and mortality.

Supplementary Appendix 1 Section 1: External validation of the NHANES mortality prediction model on the UK Biobank (UKB) dataset

We aim to validate whether the performance and explanations of the NHANES mortality prediction model generalize to an unseen population (UKB). To do so, we train a *new* tree-based 5-year mortality prediction model on the NHANES dataset using the 51 overlapping features between NHANES and UKB. As shown in Supplementary Figure 2H, the classification accuracy on the UKB test set of the model trained on NHANES samples (AUROC = 0.7780) and UKB samples (AUROC = 0.7974) are close, which shows the generalizability of the NHANES model. Supplementary Figure 1A shows the feature importances of the 51 features of the NHANES (51 features) and UKB models. *The SHAP values of both models are calculated using the same UKB samples.* We observe that the top 20 most important features are largely consistent, with 14 features the same for both models. The p-value of the Fisher's exact test (p-value = 0.0004) shows that the overlap between the top 20 most important features of both models is significant. The Spearman's correlation coefficient of both models' feature importance is 0.6969 (p-value < 0.0001). Supplementary Figures 1B-G show noteworthy results of the NHANES (51 features) model explained by UKB samples: the SHAP main effect of red cell distribution width, serum albumin and serum uric acid, and the relative 5-year mortality risk of gamma glutamyl transferase, lymphocyte percent and serum albumin. The trends shown in these figures are consistent with previous findings from both the NHANES (151 features) and UKB (51 features) models. Additional validation results on the UKB dataset are presented in Supplementary Figure 2.

Supplementary Figure 2: External validation of the NHANES mortality prediction model on the UKB dataset. (A) SHAP summary plot for the 5-year mortality prediction model trained on NHANES (51 features) dataset and explained using UKB samples. (B) The predictive performance of the models trained on the NHANES (51 features) and UKB (51 features) datasets. The AUROCs are calculated on the testing set by bootstrapping 1,000 times. (C)-(D) The main effect of serum albumin and platelet count on 5-year mortality of the model trained on the NHANES (51 features) dataset and explained using UKB samples. (E)-(F) The relative 5-year mortality risk of alanine aminotransferase ALT on male and female samples of the model trained on the NHANES (51 features) dataset and explained using UKB samples. (G) The contingency table of the Fisher's exact test that evaluates the significance of the overlap between the top 20 most important overlapping features in the model trained on the NHANES (51 features) dataset and the model trained on the UKB (51 features) dataset. Both models are explained using UKB samples.

6) In the last paragraph of the methods, the methodologic innovation needs to be better described. GBMs and Shapley plots have been extensively used in prior machine learning applications. Is the value of TreeExplainer the visualization interface? Is there another metric of explainability that is generated?

The reviewer makes an excellent point. The main contribution of this paper is utilizing explainable artificial intelligence (XAI) to do a systematic and integrated study of the associations between a large number of variables and all-cause mortality and taking a significant step towards XAI for epidemiology studies. As many examples show, a straightforward application of existing ML approaches, such as tree-based model as a predictor and the TreeExplainer as an XAI method, is far from being sufficient in

extracting knowledge that may advance the field of an application area. In this paper, we primarily focus on the application of an existing complex ML model and a state-of-the-art explanation method. In doing so, we make many important contribution, including identifying less well-studied mortality predictors, proposing an additional perspective to validate existing laboratory reference intervals, providing a new way to understand individualized mortality risk scores, and building a website for researchers and users to generate new research hypotheses and calculate mortality risk scores.

Furthermore, we make methodologic innovations: “relative risk percentage” is a new technique we propose to identify sub-optimal reference intervals, and “supervised distance” is a novel metric to measure feature redundancy and identify redundant groups of features given a specific prediction task. Building on supervised distance, we also propose a new recursive feature selection strategy to select feature sets that are both predictive and less redundant. Finally, we propose a recursive feature selection method to train accurate and efficient (low-cost) interpretable mortality risk scores.

Next, regarding the value of TreeExplainer, we would like to clarify what TreeExplainer is. TreeExplainer, which enables the exact computation of optimal local explanations for tree-based models, was proposed in our previous paper [1]. The classic Shapley values can be considered “optimal” because within a large class of approaches, they are the only way to measure feature importance while maintaining several natural properties from cooperative game theory [2]. Unfortunately, in general, computing these values exactly is NP-hard [3]. However, by focusing specifically on trees, we developed an algorithm, TreeExplainer, that computes local explanations based on the exact Shapley values in polynomial time. In the present paper, we utilize TreeExplainer to produce explanations of our tree-based mortality prediction models.

We clarify this point in *Methods Section 2.2, “IMPACT framework,” lines 117-120, lines 124-126 and lines 129-134:*

In our previous work, we introduced TreeExplainer [27], which provides a local (i.e., for each subject) explanation of the impact of input features on individual predictions for GBT models (Supplementary Methods 3). Specifically, TreeExplainer calculates exact SHAP [26] (SHapley Additive exPlanations) values for GBT models, which guarantee a set of desirable theoretical properties.

[26] Scott M Lundberg and Su-In Lee. “A unified approach to interpreting model predictions”. In: *Advances in neural information processing systems*. 2017, pp. 4765–4774.

[27] Scott M Lundberg et al. “From local explanations to global understanding with explainable AI for trees”. In: *Nature machine intelligence* 2.1 (2020), pp. 2522–5839.

In this work, we utilize TreeExplainer to conduct a systematic and integrated study of associations between a large number of variables and all-cause mortality.

In addition to studying the relationships between risk factors and all-cause mortality, we further propose a new technique, “relative risk percentage”, to identify sub-optimal reference intervals and a novel metric, “supervised distance”, to measure feature redundancy and identify redundant

groups of features given a specific prediction task. Building on supervised distance, we also propose a new recursive feature selection strategy to select feature sets that are both predictive and less redundant. We additionally propose a recursive feature selection method to train accurate and efficient (low-cost) interpretable mortality risk scores.

7) One worry I have is the individuality of predictions. For most individuals, age will be an extremely important predictor. Will the fact that age is a dominant predictor exclude the ability of Treeexplainer to identify other predictors that, while comparatively less important than age, still have considerable variability and importance between individual patients? Can the authors provide some metric of the variability of individual predictions and explainable features between patients?

We thank the reviewer for the interesting question. Although age is an important predictor, other predictors also significantly contribute to mortality prediction, as identified by TreeExplainer. As shown in Rebuttal Figures 7A below, there is high variability in mortality prediction for samples of the same age. Rebuttal Figures 7B-C also show that mortality predictions for samples of a similar age are widely spread out. This variability is caused by the contribution of other predictors. Figure 4A in the manuscript (“SHAP summary plot of the 5-year mortality prediction”) shows that the SHAP value’s variability for the top 20 most important predictors besides age is also high, which implies that a large proportion of the prediction is accounted for by these features. In terms of individualized explanations, Figures 9C-D and Supplementary Figures 4C,E illustrate that other predictors also contribute significantly to mortality prediction for these individuals. In fact, the most important predictors in Figure 9C and Supplementary Figure 4C are *not* age.

Rebuttal Figure 7: (A) Scatter plot of age vs the 5-year mortality prediction. (B) Histogram of the 5-year mortality prediction of the middle-aged samples. (C) Histogram of the 5-year mortality prediction of the older samples.

To further highlight important risk factors besides age, we can focus on explaining subpopulations of individuals who are closer in age. To do so, we define two example subpopulations: a middle-aged subpopulation (40-50) and an older subpopulation (60-70). Specifically, when we calculate the SHAP values, we explain samples in an age group by comparing them to other samples within the same age group. The SHAP summary plots are shown in Supplementary Figures 4A-B. From the figures, we observe that age is no longer the most important feature: compared to other older individuals (60-70), being older (60-70) is less remarkable. Also, as shown in Supplementary Figure 4A-B, there is no dominant predictor

because SHAP value ranges are relatively close. To highlight strong mortality predictors other than age, we find that focusing on explaining within a specific age group is quite effective.

From the individualized explanations, we also observe that other strong predictors become more important than age when explaining within a specific age group. Supplementary Figures 4C,E show the individualized explanations of a healthier and an unhealthier sample using baselines from the general population. We can see that age contributes the most to the prediction. However, as shown in Supplementary Figures 4D,F, the contribution of other important risk factors increases when we use older baselines. These examples illustrate that using the baselines with similar age can help identify the strong risk factors besides age.

We incorporate a new *Supplementary Appendix 1 Section 3, “Explaining the mortality predictions using different baseline distributions,” lines 40-58* and a new *Supplementary Figure 4*:

Supplementary Appendix 1 Section 3: Explaining the mortality predictions using different baseline distributions

In the Results section, we use TreeExplainer to explain an explicand relative to a baseline distribution drawn uniformly from all training samples (Figure 4A). This explanation substantially emphasizes age because it compares the explicand to the general population baselines that include individuals of all ages. However, in practice, epidemiologists are more interested in an individual's strong risk factors compared with people of the same age. To show this, we can manually select baselines from the samples that have similar age with the explicand. We take the middle-aged (40-50) baseline distribution and the older (60-70) baseline distribution as two examples. Specifically, we use the testing samples in the specific age range as the explicands (i.e., samples being explained) and training samples in the same age range as the baselines (i.e., background samples) when calculating the SHAP values. The SHAP summary plots are shown in Supplementary Figures 4A-B. From the figures, we observe that age is no longer the most important feature. Also, compared with Figure 4A, the SHAP value ranges are relatively similar. Therefore, we can identify the strong mortality predictors other than age for different age groups using different baseline distributions. Supplementary Figures 4C,E show the individualized explanations of a healthier vs unhealthier sample using baselines from the general population. We observe that age contributes a lot to the prediction. However, as shown in Supplementary Figures 4D,F, the contribution of other important risk factors increases when we use older baselines. These examples illustrate that using the baselines with similar age can help identify strong risk factors besides age.

Supplementary Figure 4: Explaining the 5-year mortality predictions using different baseline distributions. (A) Explaining the middle-aged subpopulation (40-50 years old) with the baselines of the same age range. (B) Explaining the older subpopulation (60-70 years old) with the baselines of the same age range. (C)-(D) The individualized explanation for an individual aged 62 using the

general population baselines and the older (60-70) baselines. (E)-(F) the individualized explanation for an individual aged 66 using the general population baselines and the older (60-70) baselines.

8) I have concerns about the conclusions about laboratory reference intervals. It is still very unclear to me how IMPACT would be used to define alternative thresholds for specific patients. Can the authors provide an analysis that suggests what optimal reference ranges are?

We thank the reviewer for highlighting this point of confusion. We note that our intention was not to argue for defining alternative thresholds for specific patients. Instead, our focus was on validating whether the reference interval agrees with our partial dependence plot analysis and identifying the examples with large disagreement for future study.

First, the relative risk percentage (RRP) is derived from *partial dependence plots*. The partial dependence function is calculated using all testing samples. Thus, it is global (for all samples), not local (for a specific sample). We do have individualized explanations (Figures 9C-D), which can suggest the strongest risk factors for a specific sample. Furthermore, relative mortality risk plots and RRP's can help to identify the reference intervals that may be sub-optimal for health. Our goal is not necessarily to suggest the optimal reference range, but instead to confirm the consistency and identify ranges that can be improved. We believe that to find optimal reference intervals, a more careful sample design and study design (e.g., randomized controlled trials) need to be established, which could be an excellent future follow-up study.

We add this to *Results Section 3.2, "Discoveries from 5-year mortality prediction," lines 239-241*:

Note that our goal is not to suggest the optimal reference range: to find the optimal reference interval, more careful sample and study design need to occur.

9) Table 2 needs more explanation. Is the RRP the relative risk of mortality for "normal" ranges compared to abnormal ranges? For example, is what this is saying that the relative risk of one-year mortality for patients with a "normal" serum albumin 28.5% that of a patient with an abnormal albumin? How does one interpret negative values here? I also have some concern with the 100% values for ALT, which are suggestive of some issue with data missingness.

Based on the reviewer's astute question, we find that the description of the relative risk percentage (RRP) in the manuscript may be insufficient. Therefore, we would like to rigorously define RRP. We use partial dependence plots to show the change in mortality risk for all values of a laboratory feature. Partial dependence plots show the marginal effect a set of features has on the prediction of an ML model. The partial function f_S is estimated by:

$$f_S(x_S) = \frac{1}{n} \sum_{i=1}^n f(x_S, x_C^{(i)}).$$

In this formula, f is an ML model, and the S are features for which the partial dependence function should be plotted. In our study, S is the laboratory feature of interest, and x_S is the given value of the feature. $x_C^{(i)}$ are actual values for features of no interest in the test set, and n is the number of instances in the test set. The partial function tells us the average marginal effect on the prediction for given values of features S . We extend the partial function to the relative mortality risk RR_S :

$$RR_S(x_S) = f_S(x_S) / \left(\frac{1}{n} \sum_{i=1}^n f(x^{(i)}) \right).$$

In other words, the relative mortality risk is defined as the average value of the model's predicted probability when we fix a specific feature to a given value divided by the average value of the model's predicted probability. We further define the relative risk percentage (RRP) as follows:

$$RRP_S = \frac{\max(RR_S(x_S), x_S \text{ in RI}) - 1}{\max(RR_S(x_S)) - 1},$$

where RI stands for reference interval.

Based on this definition, we now answer the reviewer's questions about RRP in turn. The RRP measures how large the relative mortality risk of the value within the "normal" ranges is compared to the maximum relative mortality risk of all values. The negative value means that the maximum relative mortality risk within the reference interval is smaller than 1; this indicates that the maximum average value of the model's predicted probability when we fix the laboratory feature of interest to a given value within the reference interval is smaller than the average value of the model's predicted probability. Therefore, the negative value indicates that the reference interval of that laboratory feature is optimal for mortality risk. Similarly, the 100% value means that the maximum relative mortality risk within the reference interval is equal to the maximum relative mortality risk for all values of the feature in the test set, i.e., the 100% value suggests that the reference interval may be sub-optimal for mortality risk.

We incorporated the preceding description of RRP into *the caption of Table 2* and in *Supplementary Methods Section 3.3, lines 96-110*:

Feature	Reference Interval	Relative Risk Percentage (RRP)			
		1-year	3-year	5-year	10-year
Gamma glutamyl transferase	0-30 U/L ⁴	16.93%	-4.57%	-0.97%	-6.04%
Globulin, serum	20-35 g/L ⁵	5.39%	7.95%	14.73%	4.59%
Lymphocyte percent	20%-40% ⁶	15.63%	7.02%	6.55%	10.81%
Blood urea nitrogen (Male)	2.86-8.57 mmol/L ⁷	8.12%	2.92%	8.02%	21.08%
Blood urea nitrogen (Female)	2.14-7.50 mmol/L ⁸	-0.15%	3.07%	0.40%	12.16%
Albumin, serum	35-50 g/L ⁹	28.56%	49.70%	59.77%	93.48%
Blood lead	0-0.48 umol/L ¹⁰	100.00%	94.71%	100.00%	100.00%
Mean cell volume	80-100 fL ¹¹	82.80%	75.82%	83.92%	57.26%
Alanine aminotransferase ALT (Male)	7-55 IU/L ¹²	100.00%	100.00%	100.00%	100.00%
Alanine aminotransferase ALT (Female)	7-45 IU/L ¹³	100.00%	100.00%	100.00%	100.00%

Table 2: Additional perspective to laboratory reference intervals. The table lists the reference interval and relative risk percentage (RRP; Supplementary Methods 3.3) of the selected laboratory features. The RRP measures how large the relative mortality risk of the value within the “normal” ranges is compared to the maximum relative mortality risk of all values. **A higher RRP indicates that the current reference interval is relatively more inappropriate. The negative value indicates that the reference interval of that laboratory feature is optimal for mortality risk. The 100% value suggests that the reference interval may be sub-optimal for mortality risk.**

We use *partial dependence plots* to show the change in mortality risk for all values of a laboratory feature. Partial dependence plots show the marginal effect a set of features has on the prediction of an ML model. The partial function f_S is estimated by:

$$f_S(x_S) = \frac{1}{n} \sum_{i=1}^n f(x_S, x_C^{(i)}).$$

In this formula, f is an ML model, and the S are features for which the partial dependence function should be plotted. In our study, S is the laboratory feature of interest and x_S is the given value of the feature. $x_C^{(i)}$ is actual feature values for the features of no interest in the test set, and n is the number of instances in the test set. The partial function tells us the average marginal effect on the prediction for given value(s) of features S . We extend the partial function to the relative mortality risk RR_S :

$$RR_S(x_S) = f_S(x_S) / \left(\frac{1}{n} \sum_{i=1}^n f(x^{(i)}) \right).$$

In other words, the relative mortality risk is defined as the average value of the model’s predicted probability when we fix a specific feature to a given value divided by the average value of the model’s predicted probability. We further define the relative risk percentage (RRP) as follows:

$$RRP_S = \frac{\max(RR_S(x_S), x_S \text{ in RI}) - 1}{\max(RR_S(x_S)) - 1},$$

where RI stands for reference interval.

10) For section 3.3, I would be extremely cautious about describing the relative importance of different features over different time intervals until some of the concerns around sample construction (particularly the high representation of younger individuals) are addressed. Albumin, potassium, blood pressure, etc. mean something very different for a younger rather than an older individual.

We appreciate the reviewer’s insight about this issue. As we discuss in point 3, to examine whether the large number of young samples influences the performance and explanations of the models, we subsample the younger samples and replicate key results. Specifically, we divide the samples into 20 bins based on age. Then, we subsample the samples in the first bin (aged 18-21.1) to the average number of samples in the other 18 bins except the oldest one, split the data into 80% for training and 20% for testing, and retrain the models. The new histograms of age and the relative importance of input features in the age balance 1-, 3-, 5- and 10-year mortality models are as follows:

Rebuttal Figure 8: (A)-(D) The histogram of age after the sub-sampling of young samples. (E) Relative importance of input features in the age balance 1-, 3-, 5- and 10-year mortality models.

Rebuttal Figure 8E above is generally consistent with Figure 7A in the manuscript. Also, the findings in the manuscript are also found in the age balance results: feature importance rankings change significantly between these four models. Some features are important for all four (e.g., age, RDW, and urine albumin level). Some features become more important over time (e.g., platelet count, whose importance ranking is 88 for the 1-year model and 12 for the 10-year model). Other features become less important over time (e.g., serum potassium, whose importance ranking is 21 for the 1-year model and 48 for the 10-year model). Thus, we show that the high representation of the younger sample does not have a large influence on this part of the results.

11) While I agree with temporal validation, can the authors confirm that there was no age bias in more recent datasets? E.g. in longitudinal cohort studies, healthier individuals or younger individuals may live longer, and thus the distribution in a more recent test set is not representative of the general population. Is that potentially the case here?

We thank the reviewer for raising this point. NHANES 1999-2014 is *not* a longitudinal cohort study. In that dataset, different samples are collected from different individuals. As Supplementary Figure 5 shows, the age distributions of the samples collected in different cycles are similar. Furthermore, to have similar base rates and age distribution in the test set and temporal validation set, we use the samples from different collection cycles as the temporal validation set for different follow-up times. For 1-year mortality prediction, we do a temporal validation in samples collected in 2013-2014. For 5-year mortality prediction, we do a temporal validation in samples collected in 2009-2014, and for 10-year mortality prediction, we do a temporal validation in samples collected in 2001-2014. We show the histograms of age in the training set, with the testing set and temporal validation set with different follow-up times, in Supplementary Figure 3.

We incorporate the figures in *Supplementary Figures 5 and 6*:

(A)

Follow-up time	Training set (80%) + Testing set (20%)				Temporal validation set			
	Collection cycles	Number of samples	Number of deaths	Base rate	Collection cycles	Number of samples	Number of deaths	Base rate
1-year	1999-2012	41,179	524	1.27%	2013-2014	6,082	53	0.81%
5-year	1999-2008	28,820	2,247	7.80%	2009-2014	7,034	827	11.76%
10-year	1999-2000	5,444	931	17.10%	2001-2014	16,542	4,364	26.38%

1-year mortality prediction

(B)

1999-2012

(C)

2013-2014

5-year mortality prediction

(D)

1999-2008

(E)

2009-2014

MINOR

12) In the introduction, the “notable applications of AI in healthcare” are described after the sentence around supervised learning problems. Are all the examples supervised machine learning techniques? Are any of these (e.g. image-based detection) more representative of unsupervised learning?

We thank the reviewer for this question. The three examples of “notable applications of AI in healthcare” all used supervised learning algorithms. Gulshan et al. 2016 [4] used a deep convolutional neural network for image classification, which was trained to make multiple binary predictions on diabetic retinopathy severity. Coudray et al. 2018 [5] developed a deep learning model that classified whole-slide images into normal lung, adenocarcinoma or squamous cell carcinoma and predicted the mutational status of frequently mutated genes in lung adenocarcinoma. Esteva et al. 2017 [6] trained a deep convolutional neural network for melanoma classification using dermoscopy and carcinoma classification.

Reference

- [1] S. M. Lundberg *et al.*, “From Local Explanations to Global Understanding with Explainable AI for Trees,” *Nat Mach Intell*, vol. 2, no. 1, pp. 56–67, Jan. 2020.
- [2] S. M. Lundberg and S.-I. Lee, “A unified approach to interpreting model predictions,” *Adv. Neural Inf. Process. Syst.*, vol. 30, 2017, Accessed: Apr. 06, 2022. [Online]. Available: <https://proceedings.neurips.cc/paper/2017/hash/8a20a8621978632d76c43dfd28b67767-Abstract.html>
- [3] Y. Matsui and T. Matsui, “NP-completeness for calculating power indices of weighted majority games,” *Theor. Comput. Sci.*, vol. 263, no. 1–2, pp. 305–310, Jul. 2001.
- [4] V. Gulshan *et al.*, “Development and Validation of a Deep Learning Algorithm for Detection of Diabetic Retinopathy in Retinal Fundus Photographs,” *JAMA*, vol. 316, no. 22, pp. 2402–2410, Dec. 2016.
- [5] N. Coudray *et al.*, “Classification and mutation prediction from non-small cell lung cancer histopathology images using deep learning,” *Nat. Med.*, vol. 24, no. 10, pp. 1559–1567, 2018.
- [6] A. Esteva *et al.*, “Dermatologist-level classification of skin cancer with deep neural networks,” *Nature*, vol. 542, no. 7639, pp. 115–118, Feb. 2017.

REVIEWERS' COMMENTS:

Reviewer #1 (Remarks to the Author):

Many thanks to all authors for their patient reply, I am very satisfied. Congratulations on completing such a beautiful paper.

Reviewer #2 (Remarks to the Author):

I went through the revised manuscript and authors' responses. They have responded satisfactorily . I have no more comments.

Reviewer #3 (Remarks to the Author):

No further comments